# Interventions to minimize blood loss in very preterm infants—A systematic review and meta-analysis

Emma Persad[1,2☯], Greta Sibrecht[3☯], Martin Ringsten[4], Simon Karlelid[4], Olga Romantsik[4], Tommy Ulinder[4], Israel Júnior Borges do Nascimento[5,6], Maria Björklund[7], Anneliese Arno[8], Matteo Bruschettini[4,9]*

1 Department for Evidence-Based Medicine and Evaluation, Danube University Krems, Krems an der Donau, Austria, 2 Karl Landsteiner University of Health Sciences, Krems an der Donau, Austria, 3 Poznan University of Medical Sciences, Poznan, Poland, 4 Department of Pediatrics, Lund University, Lund, Sweden, 5 University Hospital and School of Medicine, Federal University of Minas Gerais, Belo Horizonte, Minas Gerais, Brazil, 6 School of Medicine, Milwaukee Medical College of Wisconsin, Milwaukee, Wisconsin, United States of America, 7 Library & ICT, Faculty of Medicine, Lund University, Lund, Sweden, 8 Eppi-Centre, Institute of Education, University College London, London, United Kingdom, 9 Cochrane Sweden, Research and Development, Skåne University Hospital, Lund, Sweden

☯ These authors contributed equally to this work.
* matteo.bruschettini@med.lu.se

**Data Availability Statement:** All relevant data are within the manuscript and its Supporting Information files.

**Funding:** The authors received no specific funding for this work.

## Abstract

Blood loss in the first days of life has been associated with increased morbidity and mortality in very preterm infants. In this systematic review we included randomized controlled trials comparing the effects of interventions to preserve blood volume in the infant from birth, reduce the need for sampling, or limit the blood sampled. Mortality and major neurodevelopmental disabilities were the primary outcomes. Included studies underwent risk of bias-assessment and data extraction by two review authors independently. We used risk ratio or mean difference to evaluate the treatment effect and meta-analysis for pooled results. The certainty of evidence was assessed using GRADE. We included 31 trials enrolling 3,759 infants. Twenty-five trials were pooled in the comparison delayed cord clamping or cord milking vs. immediate cord clamping or no milking. Increasing placental transfusion resulted in lower mortality during the neonatal period (RR 0.51, 95% CI 0.26 to 1.00; participants = 595; trials = 5; $I^2$ = 0%, moderate certainty of evidence) and during first hospitalization (RR 0.70, 95% CI 0.51, 0.96; 10 RCTs, participants = 2,476, low certainty of evidence). The certainty of evidence was very low for the other primary outcomes of this review. The six remaining trials compared devices to monitor glucose levels (three trials), blood sampling from the umbilical cord or from the placenta vs. blood sampling from the infant (2 trials), and devices to reintroduce the blood after analysis vs. conventional blood sampling (1 trial); the certainty of evidence was rated as very low for all outcomes in these comparisons. Increasing placental transfusion at birth may reduce mortality in very preterm infants; However, extremely limited evidence is available to assess the effects of other interventions to reduce blood loss after birth. In future trials, infants could be randomized following placental transfusion to different blood saving approaches.

**Trial registration:** PROSPERO CRD42020159882.

**Competing interests:** The authors have declared that no competing interests exist.

## Introduction

### Description of the condition

The global preterm rate has been estimated to be around 11%, indicating that approximately 15 million infants are born prematurely every year worldwide [1]. The World Health Organization (WHO) defines preterm birth as a live birth occurring below the gestational age (GA) of 37 weeks [1]. The thorough classification of preterm birth is defined as the following: extremely preterm birth (GA < 28 weeks), very preterm birth (GA 28 - < 32 weeks), moderate preterm birth (GA 32 - <34 weeks), and late preterm birth (GA 34 - < 37 weeks) [1, 2]. However, it is believed that the true prevalence of premature birth is underestimated due to the gap of reporting data in several countries. Nevertheless, the rate of premature birth is reported to have increased from 9.8% in 2000 to 10.6% in 2014 [3].

The leading cause of death worldwide in children under five is attributed to complications resulting from preterm birth and the fragility of preterm infants themselves [4]. Up to 35% of deaths during the neonatal period and 16% of all deaths can be associated with preterm birth [4]. Neonatal mortality, defined as death during the initial 28 days of life, is particularly high for extremely preterm infants, and those who survive may face short- and long-term morbidities, namely bronchopulmonary dysplasia (BPD), necrotizing enterocolitis (NEC), infections, feeding difficulties and difficulties with postnatal growth, intraventricular hemorrhage (IVH), and major neurodevelopmental disabilities, such as, hearing and visual complications and behavioral and psychiatric disorders [5, 6]. The risk of the aforementioned complications increases with increasing prematurity: infants who are delivered < 28 weeks of GA have the highest mortality rate and survivors face a greater risk of developing complications due to organ immaturity. The pathogenesis of morbidities related to immaturity is heterogeneous. In this particular review, we focus on the potential influence of blood loss on common prematurity co-morbidities and interventions to prevent blood loss.

All premature infants admitted to neonatal intensive care units (NICUs) are subject to frequent blood sampling for monitoring, diagnosis, and management of potential illnesses. Blood loss due to sampling is considered a major cause of anemia in preterm infants. In fact, during the first weeks of life, extracted blood volume in critically sick neonates may be as much as 58% of the total blood volume [7–9]. In extremely premature infants, there is a speculated association between blood loss-induced anemia and both NEC [10] and BPD development [9, 11]. Anemia is usually treated with adult blood transfusions and several studies have hypothesized an association between transfusion rate and BPD, retinopathy of prematurity (ROP), and adverse neurodevelopmental outcome in extremely preterm infants [11, 12]. Although the pathogenesis of the underlying mechanisms remains unclear, one theory explaining the complications is the rapid replacement of fetal blood with adult blood, which lacks fetal blood components unique for development, namely fetal hemoglobin and circulating stems cells [13]. Further, the concentration of circulating hematopoietic stem cells has been shown to be inversely related to GA at birth [14], and there is a growing body of evidence on the differences between preterm and term mesenchymal stem cells [15, 16]. Additionally, delayed cord clamping studies [17] support the hypothesis that fetal blood components are of extreme importance in reducing the risk for immaturity-induced morbidity.

### Description of the intervention

A wide range of non-pharmacological interventions to minimize blood loss in very preterm infants has been described. Most consist of devices that avoid or reduce the need for sampling, or the amount of blood sampled, whereas delayed cord clamping and cord milking aim to

increase blood transfusion from the placenta and cord to the infant directly at birth. The following sections describe these interventions.

## Delayed cord clamping or cord milking vs. immediate clamping or no milking

The umbilical cord contains circulating fetal blood and delayed cord clamping (DCC) or umbilical cord milking (UCM) promotes an autotransfusion of fetal blood from the placenta and umbilical cord to the infant, increasing the blood volume significantly. Studies have shown beneficial effects on neonatal mortality and morbidity and favorable hemodynamic effects with more stable blood pressure [18]. A Cochrane review concluded that DCC in preterm infants <37 weeks of age may reduce mortality when compared to ICC in preterm infants [17].

There is currently no consensus on the optimal timing of cord clamping or optimal procedure of cord milking. DCC is commonly defined by cord clamping >30 seconds but can vary from 30 seconds [19] to 60 seconds [20]. The positioning of the infant relative to the placenta to promote transfusion by gravity has also been debated but does not appear to make a difference [21]. UCM is a timesaving and can be performed in critical newborns requiring resuscitation. However, it may increase risk for intraventricular hemorrhage in extremely preterm infants [22].

## Blood sampling from the cord or from the placenta vs. blood sampling from the infant

When admitting a patient to the NICU, blood samples for various analyzes is typically required. To minimize blood loss from the infant, it is possible to analyze fetal blood sampled from the umbilical cord or from the placenta. As the blood in the umbilical cord is fetal, most of the analyses that are commonly done upon admission, such as complete blood count, blood culture, blood type, antibody screening, newborn metabolic screening, and genetic testing, can be done by drawing blood from the placenta and umbilical cord rather than directly from the infant. It is even possible after DCC or UCM to draw a sufficient amount of blood from the placenta and umbilical cord. However, the accuracy for some analyses has been debated [23].

## Devices to reintroduce the blood after analysis

Blood loss could be drastically decreased through use of in-line, *ex vivo* devices where blood is withdrawn from an umbilical catheter and reintroduced via the same catheter after analysis. These tools enable the analysis of the partial pressure of oxygen ($PO_2$) and carbon dioxide ($PCO_2$), pH, HCT, and electrolytes. Standard point-of-care blood gas analysis requires between 250 to 500 microliters of blood, of which only a fraction is used for machine analysis whilst the rest is discarded. Using an in-line, *ex vivo* device, 1,500 microliters is withdrawn into a sensing chamber, analyzed, and then reintroduced, with a final blood loss of $< 25$ microliter per sample, which is a dramatic decrease in blood loss compared to standard care [24].

## Devices to monitor glucose levels, subcutaneous

Imbalance in glucose homeostasis is common in very preterm infants, where both hypo- and hyperglycemia pose challenges for clinicians. Hyperglycemia is common in very preterm infants during the first four weeks of life and associated with an increase in mortality [25] and hypoglycemia is correlated with poor neurodevelopmental outcomes [26, 27]. Continuous

glucose monitoring (CGM) has been shown to be a safe and predictable way to monitor glucose levels in pediatric and adult patients. A subcutaneous sensor connected to a transmitter on the skin measures the interstitial glucose level and sends data to a remote monitor every five minutes. The method has been validated for use in very preterm infants and appears to be well tolerated. Some studies suggest that CGM data correspond well to point-of-care glucose levels [28].

## Devices to monitor $CO_2$ levels with transcutaneous or end-tidal device

The most accurate ventilation analysis for very preterm infants is through measuring $pCO_2$ in arterial, venous, or capillary blood. However, this promotes significant iatrogenic blood loss and arterial puncture may be painful, placing a peripheral arterial catheter may be both painful and risk compromising circulation, and placing a central line may increase the risk of infection and bleeding. Non-invasive methods for monitoring ventilation, namely end-tidal $CO_2$ ($EtCO_2$) and transcutaneous $CO_2$ ($TrcCO_2$), enable continuous monitoring and decrease sampling blood loss.

$EtCO_2$ measures $CO_2$ levels in exhaled air and is widely used in pediatric intensive care units (PICUs), however is not present in NICUs due to the lack of accuracy associated with leaking from the un-cuffed tubes used for extremely preterm infants. In addition, small tidal volumes, high respiratory rate with short exhalation time, ventilation-perfusion mismatch, and the common use of high frequency oscillatory ventilation (HFOV) render this method unsuitable for very preterm infants [29].

$TrcCO_2$ measures arterial CO2 that diffuses through the skin. A sensor is placed on the skin to detect $CO_2$. By warming the sensor, a local hyperemia is created which allows for an increase of arterial blood in the capillary bed below the sensor. This method can be used regardless of the ventilation mode [29]. $TrcCO_2$ measurement has been demonstrated to be independent of birth weight, blood pressure and mean airway pressure [30] and is viewed as an accurate, complementary measure to reduce the frequency of blood sampling [31, 32].

## Devices to monitor $O_2$ levels, transcutaneous or intra-arterial vs. blood sampling

The standard method of monitoring oxygenation is pulse oximetry, which measures the percentage of hemoglobin- carrying oxygen ($satO_2$). When $satO_2$ is 100% the partial pressure of oxygen ($PaO_2$) can still increase and reach toxic levels. The standard method to measure $PaO_2$ is arterial blood sampling. Transcutaneous $pO_2$ can be measured simultaneously with $TrcCO_2$ but with less accuracy thus limiting its use in the NICU [33]. Multiparameter intra-arterial sensors are available for continuous neonatal blood gas monitoring and this method has better correlation with standard blood sample analysis, however it has not become commonly adopted in the clinical practice [34]

## Micro-methods for blood analysis

Most methods of analyzing blood samples have been developed in the infrastructure of adult medicine where the volume of blood sampled is of less importance. In very preterm infants, extensive blood sampling is one of the major contributors to neonatal anemia. Common micro-methods include dried blood spot (DBS) and volumetric absorption micro sampling (VAMS). These methods allow for sampling as little as 10–30 microliters compared to the standard amount of 500 microliters in preterm infants [35]. The implementation of micro-methods in daily practice may drastically reduce blood loss due to sampling.

## How the intervention might work

All interventions in this review have the potential to reduce blood loss in very preterm infants, thus preserving fetal blood components, such as fetal hemoglobin, stem cells and growth factors, which are essential for the prevention of morbidity and mortality in very preterm infants. Moreover, factors other than blood components might be positively influenced, such as an improvement in blood pressure [17]. Interventions such as transcutaneous or subcutaneous continuous monitoring of $CO_2$, $O_2$, and glucose might allow for the reduction in the frequency of blood sampling by real time monitoring of these values. In addition, they might lead to faster and more accurate responses to deviations from the preferred optimal values, thus preventing imbalances in ventilation, oxygenation, and glucose homeostasis, and ultimately reducing the need for blood testing.

Though strategies to preserve blood volume are unlikely to cause major harms, however there might be potential adverse events due to the specific interventions specified in this review. The lack of validation of micro methods in very preterm infants raises a question of their accuracy and might have a negative clinical impact. Transcutaneous $pCO_2/pO_2$ monitoring can leave burn marks on the skin from the sensor, which may be painful and prone to soft tissue infection. CGM relies on interstitial glucose levels measured by a subcutaneous sensor, which involves painful skin breach, increased infection risk at the CGM site, and discomfort from the transmitter on the skin. Delayed cord clamping (DCC) and umbilical cord milking (UCM) have raised concerns about having a negative effect on the infant, namely delayed resuscitation, lower Apgar scores, hypothermia, polycythemia, and hyperbilirubinemia. However, the slightly higher peak level of bilirubin in the DCC/UCM group compared to the immediate cord clamping (ICC) group did not increase the need for phototherapy [17]. Specifically for UCM, the rapid transfusion of blood may cause IVH [22].

Placental blood sampling has raised concerns about the accuracy of the analyses and whether it is possible to draw enough blood from the placenta after DCC/UCM. Several studies have investigated the validity of placental blood for admission laboratory tests and found that there is no significant difference in results compared to blood sampled directly from the infant [18].

## Why it is important to do this review

Great progress in the field of neonatology has led to an unprecedented rate of survival among preterm infants. A recent report on a Swedish national population-based cohort (2014–16) showed a survival rate of 77% in infants born between 22 and 26 gestational weeks [36]. In the same cohort, survival with any major neonatal morbidity was 62%. Anemia and blood transfusions have been associated with neonatal morbidity [9–11]. However, data from observational studies do not properly indicate whether this is only an association of poorer clinical conditions. A systematic review of randomized trials enables the assessment of a causal relationship of clinically relevant outcomes. Iatrogenic blood loss may lead to anemia and a decreased amount of endogenous circulating factors, such as fetal hemoglobin, growth factors, and hematopoietic stem cells.

We included all interventions that either avoid or reduce blood loss from birth onwards, including cord blood management, in very preterm infants.

## Methods

We included randomized or quasi-randomized controlled trials (RCTs) with parallel groups. Cluster randomized trials, cross-over randomized trials, and observational studies were not eligible for inclusion. We included studies enrolling very preterm infants (gestational age <32

weeks) admitted to the neonatal intensive care units (NICU), including the very first minutes of life in the delivery room.

We included any intervention to either avoid or reduce blood loss and their respective comparators:

- Delayed cord clamping or cord milking vs. immediate cord clamping or no cord milking: delayed being defined as after 30 seconds and 'immediate' as within 30 seconds;

- Blood sampling from the umbilical cord or from the placenta vs. blood sampling from the infant;

- Devices to reintroduce the blood after analysis vs. conventional blood sampling;

- Devices to monitor glucose levels, subcutaneous vs. conventional blood sampling;

- Devices to monitor $CO_2$ levels with transcutaneous or end-tidal device vs. measurement of $CO_2$ levels measured by conventional blood sampling;

- Transcutaneous or intra-arterial measurement of $O_2$ levels to avoid/reducing blood sampling vs. measurement of $O_2$ levels by conventional blood sampling;

- Micro-methods to use up to 20 μL vs. methods to use >20 μL.

The primary outcomes were:

- All-cause neonatal mortality (first 28 days of life);

- All-cause mortality during initial hospitalization;

- One-year survival;

- Major neurodevelopmental disability: cerebral palsy, developmental delay (Bayley Mental Developmental Index [37, 38] or Griffiths Mental Development Scale [39] assessment > 2 SDs below the mean), intellectual impairment (IQ > 2 SDs below the mean), blindness (vision < 6/60 in both eyes), or sensorineural deafness requiring amplification [40] *for children 18 to 24 months*;

- Major neurodevelopmental disability: cerebral palsy, developmental delay (Bayley Mental Developmental Index [37, 38] or Griffiths Mental Development Scale [39] assessment > 2 SDs below the mean), intellectual impairment (IQ > 2 SDs below the mean), blindness (vision < 6/60 in both eyes), or sensorineural deafness requiring amplification [40] *for children 3 to 5 years of age*;

- Mortality or major neurodevelopmental disability [composite outcome].

The secondary outcomes were:

- Any ROP;

- Severe ROP ($\geq$ stage 3) [41];

- Any germinal matrix-IVH grades 1 to 4 (according to Papile classification [42];

- Severe IVH: ultrasound diagnosis grades 3 or 4 (according to Papile classification [42];

- White matter at term-equivalent MRI abnormalities at term equivalent age (yes/no), defined as white matter lesions (i.e. cavitations; [43], punctate lesions [44], GM-IVH [45], or cerebellar hemorrhage [46];

- BPD/chronic lung disease: 28 days [47]; 36 weeks' postmenstrual age [48]; 'physiological definition' [49];

- NEC (defined as Bell's $\geq$ stage II) [50];

- Volume in mL of blood withdrawn until hospital discharge;

- Volume in mL of blood transfused until hospital discharge;

- Number of blood transfusions until hospital discharge;

- Need for blood transfusions until hospital discharge;

- Concentrations of total hemoglobin (Hb);

  ○ Day 1 to 7 of life;

  ○ Day 8 to 14 of life;

  ○ After 15 days of life.

- Concentrations of fetal hemoglobin (Hb F);

  ○ Day 1 to 7 of life;

  ○ Day 8 to 14 of life;

  ○ After 15 days of life;

- Late sepsis until hospital discharge;

- PDA (pharmacological treatment and surgical ligation);

- Duration in days of respiratory support (i.e. nasal continuous airway pressure and ventilation via an endotracheal tube considered separately and in total);

- Duration in days of supplemental oxygen requirement;

- Duration in days of hospital stay;

- Each component of the composite outcome 'major neurodevelopmental disability' (see Primary outcomes);

- Poor academic performance at 12 years of age defined as one or more standard scores of less than 70 (2 SD below the mean of 100) on the four subtests of the Wide Range Achievement Test–4: sentence comprehension, word reading, spelling, and math computation [51];

- Motor skills (manual dexterity, aiming and catching, and balance) measured with the Movement Assessment Battery for Children–Second Edition (Movement ABC-2) [52], in which total standard scores corresponding to the 5th percentile or less were defined as motor impairment.;

- Behavioral problem defined as a Total Problem T score of greater than 69 ($\geq$2 SD above the mean of 50) on the Child Behavior Checklist that was completed by the primary caregiver [53];

- Time in minutes to perform the procedure in each study arm and within each comparison (not for delayed cord clamping);

- Pain during device insertion/use and blood sampling, e.g. heel stick, venipuncture;

- Number of skin-breaking procedures associated to blood testing, insertion and repositioning of the device; intermittent modalities to measure glycemia (e.g. capillary glucose testing; venipuncture);

- Skin/soft tissue injury associated to blood testing, insertion and repositioning of the device; intermittent modalities to measure glycemia (e.g. capillary glucose testing; venipuncture);

- Site infection associated to blood testing, insertion and repositioning of the device; intermittent modalities to measure glycemia (e.g. capillary glucose testing; venipuncture);

- Thrombotic event rates

We conducted a comprehensive, systematic literature review search in the following databases: The Cochrane Central Register of Controlled Trials (CENTRAL, The Cochrane Library, issue 12, 2019), PubMed (MEDLINE, 1966 to December 2019), EMBASE (Elsevier, 1988 to December 2019), CINAHL Complete (Ebsco), and LILACS (Latin American and Caribbean Health Sciences Literature) for eligible studies to be included. No language or publication date restrictions were applied.

The search strategy for each database was designed by an information specialist and librarian at Lund University (see S1 File for full search strategy). The primary search was conducted on 10 December 2019, after test search modifications were made to better identify relevant studies by splitting the search strategy into strands based on intervention. A secondary search was performed on 11 January 2020 after an additional intervention was added, blood sampling from the placenta vs. from the infant.

We searched https://clinicaltrials.gov and WHO International Clinical Trials Registry Portal, ICTRP, for ongoing trials and cross-checked the references of relevant background literature and systematic reviews to ensure we had captured all relevant titles.

Once the search process was complete, all matching references were imported into Covidence–a web-based program used for screening and data extraction in systematic reviews [54]. To assess whether a study should be included, study design, types of participants, interventions and comparators as well as pre-specified exclusion criteria were considered.

Two review authors independently screened each reference for relevant studies to be included. Initially, references were screened using titles and abstracts. Following that, potentially relevant studies were screened once again in full text-format. We resolved any disagreements by discussion and, if necessary, by consulting a third reviewer.

Two reviewers independently extracted data from the included studies using the built-in extraction form in Covidence extracting the following:

- Identification–sponsorship source; country; setting; authors name; contact details;

- Methods–Study design; study grouping;

- Participants–Baseline characteristics for the two groups: gestational age; birth weight; number of patients at randomization and at outcome. Inclusion criteria; exclusion criteria;

- Interventions–characteristics with description of procedure in each group.

Outcomes were extracted into a custom-made review template created according to our pre- specified outcomes from the protocol, ensuring all outcomes were extracted identical in spelling and no outcomes were missed. Study authors of included studies were contacted for further data on methods and/or results when needed. Studies with multiple references or publications were merged in Covidence under the same study ID as the main study, named after the first author, followed by publication year of the main study.

All discrepancies in data extracted between the two reviewers were identified automatically in Covidence, requiring intervention before a consensus could be completed. These discrepancies were solved primarily by discussion between the two reviewers and ultimately by a third review author.

The methodological quality of the individual studies was assessed by two review authors independently using the Cochrane risk of bias tool [55] which encompasses the following domains:

1. Random sequence generation: selection bias due to inadequate generation of a randomized sequence;

2. Allocation concealment: selection bias due to inadequate concealment of allocations prior to assignment;

3. Blinding of participants and personnel: performance bias due to knowledge of the allocated interventions by participants and personnel during the study;

4. Blinding of outcome assessment: detection bias due to knowledge of the allocated interventions by outcome assessors;

5. Incomplete outcome data: attrition bias due to amount, nature or handling of incomplete outcome data;

6. Selective reporting: reporting bias due to selective outcome reporting;

7. Other bias: bias due to problems not covered elsewhere.

A risk of bias model and a summary table were used to illustrate risk across studies and of individual studies. Any disagreements among the authors were solved by discussion and ultimately though involving a third reviewer. We conducted measures of treatment effect data analysis using the Cochrane software Review Manager 5.4 [56]. We determined outcome measures for dichotomous data (e.g. death, frequency of intraventricular hemorrhage) as risk ratios (RRs) with 95% confidence intervals (CIs). We calculated continuous data (e.g. amount of blood loss, duration of respiratory support) using mean differences (MDs) and SDs. We contacted authors to request missing data when needed. We used the $I^2$ statistic with the following cut-offs for heterogeneity: less than 25% no heterogeneity; 25% to 49% low heterogeneity; 50% to 74% moderate heterogeneity; and $\geq$ 75% high heterogeneity (Higgins 2003). We investigated reporting bias using funnel plots for primary outcomes of high clinical relevance with a high number of included studies. We assessed the asymmetry of the funnel plots visually. If asymmetry was present, we planned to investigate it. Most outcomes were not reported on by enough studies to perform these analyses. Data were analyzed on an intention-to-treat basis and with a fixed-effect model. We synthesized the data with risk ratio (RR) for dichotomous outcomes and mean difference (MD) for continuous outcomes, with 95% confidence intervals (CI). When meta-analysis was judged to be inappropriate, we analyzed and interpreted the individual studies separately. If two or more studies reported an outcome the results were pooled. The certainty of evidence was assessed through the Grading of Recommendations Assessment, Development and Evaluation (GRADE) approach, as outlined in the GRADE Handbook [57] for the primary outcomes. This was done by rating limitations in study design or execution, rating inconsistency and imprecision in results as well as rating the indirectness of evidence and publication bias–ultimately yielding a classification of the certainty of evidence in one of four grades: high, moderate, low or very low. We planned to perform a subgroup analysis for premature infants of gestational age fewer than 28 weeks; premature infants of gestational age equal or greater than 28 weeks and fewer than 32 weeks.

## Results

### Description of studies

**Results of the search.** A total of 10,399 references were obtained following the literature search. After removing 1,745 duplicates, 8,644 references remained for title and abstract screening. After screening, 8,448 references did not match our inclusion criteria, leaving 196 full-text references to be assessed for eligibility (Fig 1). Eighty references were deemed ineligible and excluded. Forty-nine references were found to be a duplicate or a subset of another study and were merged into one study. A grand total of 67 studies were ultimately included in the full review, with 23 ongoing trials, 13 awaiting classification, and 31 regular studies. We found no retractions, withdrawals or questions on research quality for any of the included studies in the retraction watch database (http://retractiondatabase.org/). We have listed characteristics of populations and interventions and comparisons of the 31 included studies in Table 1 and S2 File. Twenty-five studies concerned the intervention delayed cord clamping or cord milking vs. immediate cord clamping or no milking. Two studies were relevant for the comparison blood sampling from the umbilical cord or from the placenta vs. blood sampling from the infant, one study was relevant for the comparison devices to reintroduce the blood after analysis vs. conventional blood sampling, and three studies were relevant for the comparison devices to monitor glucose levels, subcutaneous vs. conventional blood sampling. There were no relevant studies included for remaining interventions micro-methods to use up to 20 μL vs. methods to use >20 μL, devices to monitor $CO_2$ levels with transcutaneous or end-tidal device vs. measurement of $CO_2$ levels measured by conventional blood sampling, and transcutaneous or intra-arterial measurement of $O_2$ levels to avoid/reducing blood sampling vs. measurement of $O_2$ levels by conventional blood sampling.

### Included studies

We analyzed data from 31 studies that recruited 3,759 infants: 3,421 infants from 25 studies comparing delayed cord clamping or cord milking vs. immediate cord clamping or no milking [58–82], 124 infants from two studies comparing blood sampling from the umbilical cord or from the placenta vs. blood sampling from the infant [83, 84], 121 infants from three studies comparing devices to monitor glucose levels, subcutaneous vs. conventional blood sampling [85–87], and 93 infants from one study comparing devices to reintroduce the blood after analysis vs. conventional blood sampling [24].

Three of the studies on cord management were multicenter [24, 79, 82]. The included studies were conducted in 14 countries: nine in the United States [59, 65, 70, 71, 74–77, 84]; three in Turkey [58, 68, 81], two in Canada [61, 66], in India [62, 83], in Switzerland [60, 78], and in the United Kingdom [64, 86]; one in China [63], in France [87], in Germany [80], in Iran [72], in Israel [73], in Ireland [67], in Italy [85], in Japan [69]. Tables 1 and 2 show additional information on populations and characteristics of the interventions.

### Excluded studies, ongoing studies and studies awaiting classification

Following full-text screening, 80 studies were excluded for the following reasons: infants > 32 weeks (21 studies); wrong comparator, e.g. no immediate cord clamping (21 studies); wrong study design, e.g. not an RCT (15 studies); duplicate study (11 studies); wrong intervention, e.g. delayed cord clamping done before 30 sec (7 studies); terminated before completion (3 studies); not enough information (2 studies). The reasons for exclusion of each study are listed in S3 File

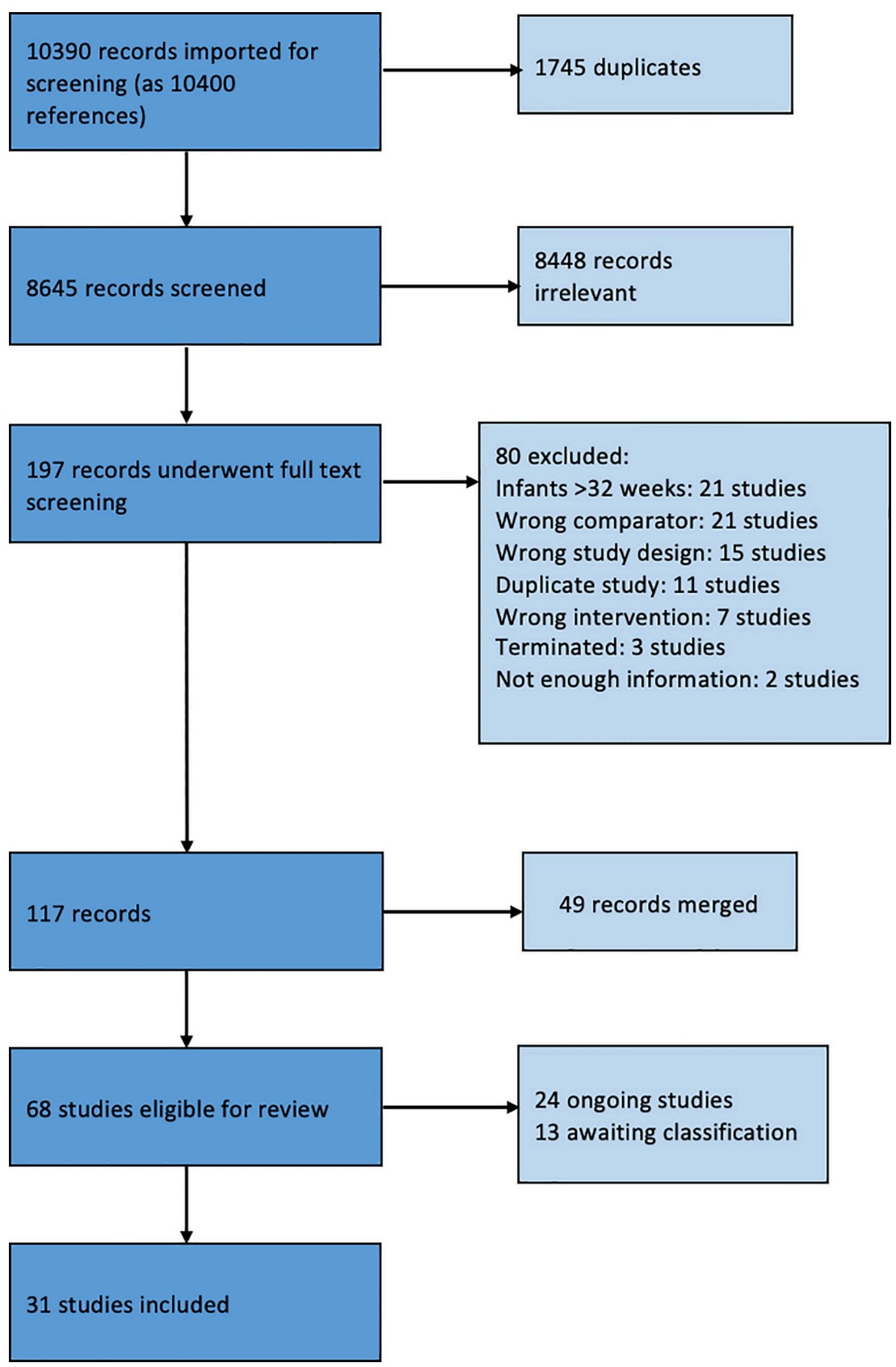

**Fig 1. PRISMA flow diagram.**

**Table 1. Characteristics of populations, interventions, and comparisons of included studies.**

| Study ID (no. infants) | Country | Gestational age, weeks | | Intervention | Comparison | Co-interventions | Other Notes |
|---|---|---|---|---|---|---|---|
| | | Intervention | Comparison | | | | |
| **1a) Delayed vs. immediate cord clamping** | | | | | | | |
| Aladangady 2006 (46) | Scotland | Not reported (shows numbers of infants in different subgroups) | Not reported (shows numbers of infants in different subgroups) | Median at 90 sec | Median at 10 sec | | |
| Duley 2018 (276) | United Kingdom | Median 29 (IQR 27.1 to 30.7) | 29.1 (IQR 27.6 to 30.4) | ≥2 min per protocol | ≤20 sec per protocol | | |
| Backes 2016 (40) | USA | Mean 24.4 (SD 1.2) | Mean 24.6 (SD 1.1) | Mean at 37.4 sec | Mean at 3.8 sec | | |
| Baenziger 2007 (39) | Switzerland | Mean 30.43 (SD 2.3) | Mean 29.71 (SD 2.38) | Between 60 to 90 seconds per protocol | <20 sec per protocol | "The delivery of the infants in the experiment group was immediately followed by maternal administration of syntocinon" | |
| Chu 2019 (38) | Canada | Mean 26.4 (SD 0.9) | Mean 29.4 (SD 2.0) | 39.7 ± 36.2 sec | 5.4 ± 5.0 sec | | |
| Dong 2016 (90) | China | Mean 29.5 (SD 1.7) | Mean 29.1 (SD 1.6) | At 45 sec per protocol | <10 sec per protocol | | |
| Kazemi 2017 (70) | Iran | Mean 30.1(SD 1.7) | Mean 29.8 (SD 1.8) | 30–45 sec per protocol | <10 sec per protocol | | |
| Nelle 2012 (35) | Switzerland | Mean 29.0 (SD 2) | 28.6 (SD 2) | >30 sec per protocol | Not reported | | |
| Oh 2011 (33) | USA | Mean 26 (SD 1.4) | Mean 26 (SD 1.1) | 35.2 ±10.1 sec | 7.9 ± 5.2 sec | | |
| Rabe (40) 2000 | Germany | Mean 30.01 (SD 1.57) | Mean 29.48 (SD 1.96) | 45 sec per protocol | 20 sec per protocol | | |
| Tarnow-Mordi 2018 (1634) | Australia | Mean 28 (SD 2) | Mean 28 (SD 2) | Median 60 sec | Median 5 sec | | |
| Mercer 2003 (32) | USA | Mean 28 (SD 2) | Mean 27 (SD 2.2) | 32±12 sec as measured | 6.2±3 sec as measured | No uterotonics were given before the cord clamping. | |
| Gokmen 2011 (42) | Turkey | Mean 29.3 (SD 1.2) | Mean 29.4 (SD 1.5) | 30–45 sec per protocol | <10 sec per protocol | No uterotonics were given before cord clamping | |
| Dipak 2017 (53) | India | Mean 30.1 (SD 1.2) | Mean 29.9 (SD 1.4) | at 60 sec per protocol | at 10 sec per protocol | | We included only the DCC1 group, as in DCC2 group ergometrine (500 mcg i.m) was given to the mother before cord clamping. |
| Kugelman 2007 (36) | Israel | Mean 30.3 (SD 1.78) | Mean 29.8 (SD 1.95) | 30–45 sec per protocol | < 10 sec per protocol | No uterotonics were given before cord clamping. | |
| Elimian 2014 (200) | USA | Mean 30.9 (SD 3.1) | Mean 30.7 (SD 2.8) | Median 32 sec (range 31–35 sec) as measured | Median 2 sec (range 1–5 sec) as measured | Three to four passes of milking of the umbilical cord toward the neonate were allowed in all neonates in the DCC group. Oxytocin infusion was not initiated during the period of cord clamping and was started only after the delivery of the placenta. | |

*(Continued)*

**Table 1.** (Continued)

| Study ID (no. infants) | Country | Gestational age, weeks | | Intervention | Comparison | Co-interventions | Other Notes |
|---|---|---|---|---|---|---|---|
| | | Intervention | Comparison | | | | |
| Mercer 2016 (208) | USA | Mean 28.3 (SD 2) | Mean 28.4 (SD 2) | 32 +- 16 sec as measured | 6.6 +- 6 sec as measured | At 30 to 45 seconds the obstetrician was asked to milk the infant's cord once then clamp and cut the umbilical cord. | |
| *Finn 2019 (26 ICC+DCC) | Ireland | Median 28 (IQR 26.4–29.6) | Median 28.5 (IQR 25.7–30.5) | at 60 sec per protocol | <20 sec per protocol | Routine neonatal care was provided, including positive end-respiratory pressure and the provision of positive pressure ventilation if required. | |
| Mercer 2006 (72) | USA | Mean 28.3 (SD 2.1) | Mean 28.2 (SD 2.4) | Mean 41 (SD 9) as measured | Mean 5 (SD 7) as measured | | |
| **1b) Milking vs. no milking** | | | | | | | |
| *Finn 2019 (31 ICC +UCM) | Ireland | Median 28.4 (IQR 25.7–29.6) | Median 28.5 (IQR 25.7–30.5) | Milked 3 times, 20 cm over 2 seconds | <20 sec per protocol | | |
| Alan 2014 (48) | Turkey | Mean 28.4 (SD 1.8) | Mean 28.0 (SD 1.9) | Milked 3 times. Approx. 5 cm/s | ICC at <10 sec | | |
| March 2013 (113) | USA | Median 27.0 (IQR 25.5 to 28.1) | Median 26.3 (IQR 25.1 to 27.1) | Milked 3 times. No speed reported. | ICC. No time reported. | | |
| El-Naggar 2019 (73) | Canada | Mean 27.6 (SD 1.8) | Mean 27.2 (SD 2.0) | Milked 3 times. Approx. 10cm/s | ICC at <10 sec | | |
| Hosono 2008 (40) | Japan | Mean 27.0 (SD 1.5) | Mean 26.6 (SD 1.2) | Milked 2–3 times. Approx 10 cm/s | ICC. No time reported. | | |
| Josephsen 2014 (26) | USA | Mean 26.5 (SD 1.4) | Mean 26.1 (SD 0.9) | Milked 3 times. Speed not reported. | ICC. No time reported. | | |
| Katheria 2014 (60) | USA | Mean 28 (SD 2) | Mean 28 (SD 2) | Milked 3 times. Approx 10 cm/s | ICC at mean 14 sec (SD 9). | | |
| Silahli (75) 2018 | Turkey | <32 gestational weeks | <32 gestational weeks | 20 cm of umbilical cord was milked 3 times | < 10 sec per protocol | | |
| **2) Blood Sampling from the Umbilical Cord or Placenta versus Blood Sampling from the Infant** | | | | | | | |
| Prescott 2016 (44) | USA | Mean 28 (SD 3) | Mean 27 (SD 3) | Sampling at admission from the umbilical cord | Sampling at admission from the infant | | |
| Balasubramanian 2019 (71) | India | Mean 26.5 (SD 1.3) | Mean 26.4 (SD 1.4) | Initial cord milking followed by 5 mL of blood drawn from the umbilical vain. | Initial cord milking followed by 5 mL of blood drawn from the umbilical vain which was then discarded. 5 mL of blood was then collected from the infant within the first hour of life. | | |
| **3) Reintroduction of blood after analysis vs. no reintroduction of blood after analysis** | | | | | | | |
| Widness 2005 (83) | USA | Mean 26.0 (SD 2.0) | Mean 26.0 (SD 1.8) | Withdrawl of 1.5 ml blood samples through an umbilical artery catheter for analysis and reintroduces all but 25 μL of the withdrawn blood | Routine care with laboratory blood analysis. | | |

*(Continued)*

**Table 1.** (Continued)

| Study ID (no. infants) | Country | Gestational age, weeks | | Intervention | Comparison | Co-interventions | Other Notes |
|---|---|---|---|---|---|---|---|
| | | Intervention | Comparison | | | | |
| **4) Continuous glucose monitoring vs. no continuous glucose monitoring** | | | | | | | |
| Galderisi 2017 (50) | Italy | Median 30 (IQR 29–31) | Median 30 (IQR 28–31) | CGM device with active alarms with threshold values at <47 and >180 mg/dL | CGM device with blinded monitor and no alarm. Point of care blood glucose testing occurred minimum every 8 hours and in presence of clinically relevant events. | | |
| Uetwiller 2015 (43) | France | Median 30.1 (Min 24.4, max 37.0) | Median 29.6 (Min 24.1, max 34.7) | CGM device with active alarms with a threshold value at <60 mg/dL followed by a capillary blood test. | Point of care capillary blood glucose testing every 4 hours. | | |
| Thomson 2005 (20) | UK | Mean 27.5 (SD 2.8) | Mean 28.0 (SD 2.1) | CGM device where blood glucose was registered every hour by a nurse | CGM device with blinded data together with standard care | | |

\* = This trial is described in both a (delayed vs. immediate cord clamping) and b (milking vs. no milking) as it comprised three arms: delayed cord clamping; cord milking; and immediate cord clamping without milking.

We identified 24 ongoing studies within the following comparisons: delayed cord clamping, or cord milking compared to immediate clamping or no milking (22 studies); devices to monitor glucose levels (1 study); Micro-methods for blood analysis (1 study). Full description of each ongoing study is reported in S4 File.

We identified 13 studies awaiting classification, all on cord clamping management. Full description of each of these studies is reported in S5 File.

### Risk of bias in included studies

We have presented a summary of the 'Risk of bias' assessment in Figs 2 and 3. We have provided details of the methodological quality of included trials in S2 File.

**Allocation (selection bias).** Although the included studies all contained a statement about randomization, this often remained unclear.

Seventeen studies described the generation of the randomization sequence and were at low risk of bias in this subsection [24, 59, 62, 64–67, 69, 74, 76, 81–87]. Fourteen studies did not mention the generation of the randomization sequence and were at unclear risk of bias in this subsection [17, 29, 58, 60, 61, 63, 68, 70–72, 75, 77–79].

Twenty-three studies described a strategy for concealment of the allocation sequence and were at low risk of bias in this subsection [24, 58, 59, 61, 62, 64–67, 69, 71, 73–76, 80–87]. Eight studies did not mention the concealment of the allocation sequence and were at unclear risk of bias in this subsection [60, 63, 68, 70, 72, 77–79].

**Blinding (performance bias and detection bias).** All trials were either reported as unblinded or did not mention blinding of participants and personnel (performance bias), leading to high risk of performance bias. However, it should be considered that blinding does not affect mortality, limiting the risk of dealing with biased findings for this outcome.

**Table 2. Summary of findings table for comparison: 1 delayed cord clamping or milking vs. immediate cord clamping or no milking.**

Delayed cord clamping or milking compared to immediate cord clamping or no milking for minimizing blood loss

**Patient or population**: Very preterm infants

**Setting**: Clinical

**Intervention**: Delayed cord clamping or milking

**Comparison**: Immediate cord clamping or no milking

| Outcomes | № of participants (studies) Follow up | Certainty of the evidence (GRADE) | Relative effect (95% CI) | Anticipated absolute effects | |
|---|---|---|---|---|---|
| | | | | Risk with immediate cord clamping or no milking | Risk difference with Delayed cord clamping or milking |
| All-cause neonatal mortality (first 28 days) | 734 | ⊕⊕⊕◯ | RR 0.50 | 80 per 1,000 | **40 fewer per 1,000** |
| | (7 RCTs) | MODERATE [a] | (0.27 to 0.92) | | (58 fewer to 6 fewer) |
| All-cause mortality during initial hospitalization | 2476 | ⊕⊕◯◯ | RR 0.70 | 71 per 1,000 | **21 fewer per 1,000** |
| | (10 RCTs) | LOW [a,b] | (0.51 to 0.96) | | (35 fewer to 3 fewer) |
| Cognitive disability for children alive at 24 months of age | 276 | ⊕◯◯◯ | RR 1.86 | 43 per 1,000 | **37 more per 1,000** |
| | (1 RCT) | VERY LOW [c,d] | (0.71 to 4.89) | | (13 fewer to 168 more) |
| Cognitive disability for children alive at 3 years of age | 73 | ⊕◯◯◯ | RR 0.14 | 83 per 1,000 | **72 fewer per 1,000** |
| | (1 RCT) | VERY LOW [c,d] | (0.01 to 2.27) | | (82 fewer to 106 more) |
| Death or adverse neurodevelopmental outcome for children 18 to 24 months of age | 276 | ⊕◯◯◯ | RR 0.70 | 252 per 1,000 | **76 fewer per 1,000** |
| | (1 RCT) | VERY LOW [c,d,e] | (0.44 to 1.11) | | (141 fewer to 28 more) |
| One year survival | (0 RCTs) | - | not estimable | 0 per 1,000 | **0 fewer per 1,000** |
| | | | | | (0 fewer to 0 fewer) |

*The risk in the intervention group** (and its 95% confidence interval) is based on the assumed risk in the comparison group and the **relative effect** of the intervention (and its 95% CI).

**CI:** Confidence interval; **RR:** Risk ratio.

**GRADE Working Group grades of evidence.**

**High certainty:** We are very confident that the true effect lies close to that of the estimate of the effect

**Moderate certainty:** We are moderately confident in the effect estimate: The true effect is likely to be close to the estimate of the effect, but there is a possibility that it is substantially different.

**Low certainty:** Our confidence in the effect estimate is limited: The true effect may be substantially different from the estimate of the effect.

**Very low certainty:** We have very little confidence in the effect estimate: The true effect is likely to be substantially different from the estimate of effect.

**Explanations**

a. Downgraded for imprecision by one level: Few events, wide CIs.

b. Large heterogeneity between trials for the frequency of events for this outcome.

c. Downgraded for only including one study.

d. Downgraded due to few events and wide CI.

e. Downgraded one level for no blinding of personnel, uncertain blinding for outcome assessors and for a significant difference between lost in follow up between groups.

Six studies mentioned blinding in the blinding of outcome assessment (detection bias) subsection [59, 61, 68, 74, 77, 83] and were rated at a low risk of bias in this subsection. Ten studies were rated at an unclear risk of bias in this subsection [64, 66,71, 75, 76, 80, 81, 84, 85, 87]. All other studies were rated at high risk of bias.

**Incomplete outcome data (attrition bias).** Twenty-two studies explained and accounted for exclusions, drop-outs, and changes in outcome reporting and were rated at low risk of bias [59, 61–69, 71–77, 80, 82, 83, 85, 87].

Six studies only partially explained exclusions, drop-outs, and changes in outcome reporting and were rated at unclear risk of bias [58, 60, 70, 78, 79, 81].

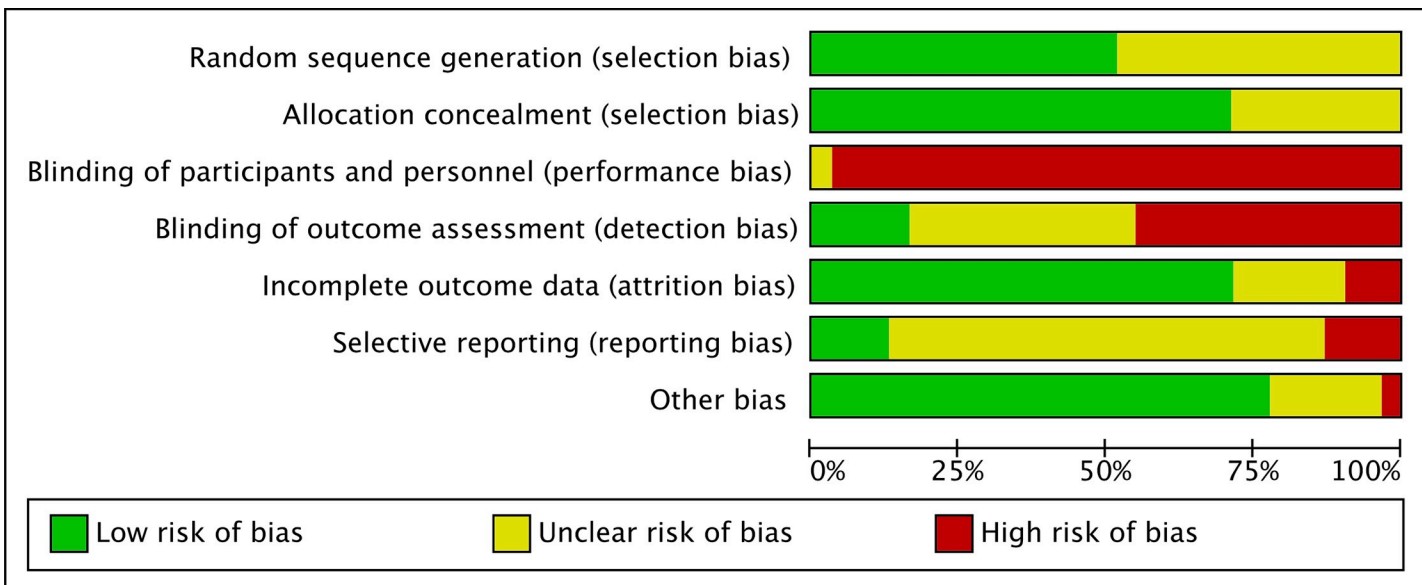

**Fig 2. Methodological quality graph.** Authors' judgements about each methodological quality item presented as percentages across all included studies.

Three studies did not explain exclusions, drop-outs, and changes in outcome reporting and were rated at high risk of bias [24, 84, 86].

**Selective reporting (reporting bias).**   When available, we compared the protocol or a trial registration with the publication and could not detect selective reporting in four studies, which we rated at a low risk of bias [64–66, 82]. In 23 studies, no protocol or trial registry entry was available so selective reporting bias remains unclear [24, 58–61, 63, 68–71, 73–81, 83, 85–87]. In four studies, there were unclear changes to the protocol and the studies were rated at high risk of bias [62, 67, 72, 84].

**Other potential sources of bias.**   Other potential sources of bias were not found in most studies.

Six studies were rated at unclear risk of bias [60, 61, 65, 85–87]. Two studies had uneven distribution between the groups [60, 65], two studies had unevenly distributed maternal characteristics which may have influenced the results [86, 87], one did not report units properly [61], and one had differences in baseline characteristics [85]. One study is an ongoing trial and only reported interim data [84]. It was rated at a high risk of bias.

## Effects of interventions

**Comparison 1: Delayed cord clamping or cord milking vs. immediate cord clamping or no milking.**   Twenty-five studies with 3,421 infants compared delayed cord clamping or cord milking compared to immediate cord clamping or no milking [58–82]. Seventeen studies compared delayed cord clamping to immediate cord clamping [10, 60–65, 67, 68, 72, 73, 75–80, 82]. Eight studies compared cord milking to immediate cord clamping [58, 66, 67, 69–71, 74, 81].

The Summary of Findings Table shows the certainty of evidence for the primary outcomes (Table 2).

*1.1 All-cause neonatal mortality (first 28 days).* Seven studies reported on neonatal mortality (first 28 days); five studies comparing delayed vs. immediate clamping [59, 60, 64, 65, 80] and two studies comparing milking vs. immediate cord clamping [70, 74]. The intervention was

Risk of bias summary (+ = low risk, ? = unclear risk, − = high risk):

| Study | Random sequence generation (selection bias) | Allocation concealment (selection bias) | Blinding of participants and personnel (performance bias) | Blinding of outcome assessment (detection bias) | Incomplete outcome data (attrition bias) | Selective reporting (reporting bias) | Other bias |
|---|---|---|---|---|---|---|---|
| Alan 2014 | ? | + | − | − | ? | ? | + |
| Backes 2016 | + | + | − | + | + | ? | + |
| Baenziger 2007 | ? | ? | − | − | ? | ? | ? |
| Balasubramanian 2019 | + | + | − | + | + | ? | + |
| Chu 2019 | ? | + | − | + | + | ? | ? |
| Dipak 2017 | + | + | − | − | + | − | + |
| Dong 2016 | ? | ? | − | − | + | ? | + |
| Duley 2018 | + | + | − | ? | + | + | + |
| Elimian 2014 | + | + | − | − | + | + | ? |
| El Naggar 2019 | + | + | − | ? | + | + | + |
| Finn 2019 | + | + | − | − | + | − | + |
| Galderisi 2017 | + | + | − | ? | + | ? | ? |
| Gokmen 2011 | ? | ? | − | + | + | ? | + |
| Hosono 2008 | + | + | − | − | + | ? | + |
| Josephsen 2014 | ? | ? | − | − | ? | ? | + |
| Katheria 2014 | ? | + | − | ? | + | ? | + |
| Kazemi 2017 | ? | ? | − | − | + | − | + |
| Kugelman 2007 | ? | + | − | − | + | ? | + |
| March 2013 | + | + | − | + | + | ? | + |
| Mercer 2003 | ? | + | − | ? | + | ? | + |
| Mercer 2006 | + | + | − | ? | + | ? | + |
| Mercer 2016 | ? | ? | − | + | + | ? | + |
| Nelle 2012 | ? | ? | − | − | ? | ? | + |
| Oh 2011 | ? | ? | − | − | ? | ? | + |
| Prescott 2014 | + | + | − | ? | − | − | − |
| Rabe 2000 | ? | + | − | ? | + | ? | + |
| Silahli 2018 | + | + | − | ? | ? | ? | + |
| Tarnow Mordi 2018 | + | + | − | − | + | + | + |
| Thomson 2018 | + | ? | − | − | − | ? | ? |
| Uettwiller 2015 | + | + | − | ? | + | ? | ? |
| Widness 2005 | + | + | − | − | − | ? | + |

**Fig 3. Methodological quality summary.** Authors' judgements about each methodological quality item for each included study.

associated with lower mortality rates (4% vs. 8%) and the confidence interval included a potential beneficial effect (RR 0.50, 95% CI 0.27, 0.92; 7 RCTs, participants = 734, moderate certainty of evidence).

*1.1.1 Delayed vs. immediate clamping*

Five studies reported on neonatal mortality (first 28 days) in the delayed vs. immediate clamping subgroup [59, 60, 64, 65, 80]. We are uncertain of the effect of delayed cord clamping compared to immediate clamping on the rates of all-cause neonatal mortality rates (first 28 days) (RR 0.51, 95% CI 0.26 to 1.00; participants = 595; studies = 5; $I^2$ = 0%). This suggests that if the all-cause neonatal mortality rate (first 28 days) following immediate clamping is 8%, the rate of all-cause neonatal mortality (first 28 days) following delayed cord clamping would be between 2% and 8%.

*1.1.2 Milking vs immediate clamping*

Two studies reported on neonatal mortality (first 28 days) in the milking vs. immediate clamping subgroup [70, 74]. We are uncertain of the effect of milking compared to immediate clamping on the rates of all-cause neonatal mortality rates (first 28 days) (RR 0.46, 95% CI 0.11 to 2.00; participants = 139; studies = 2; $I^2$ = 0%). This suggests that if the all-cause neonatal mortality rate (first 28 days) following immediate clamping is 7%, the rate of all-cause neonatal mortality (first 28 days) following milking would be between 1% and 14%.

*1.2 All-cause mortality during initial hospitalization.* Ten studies reported on neonatal mortality during initial hospitalization; seven studies comparing delayed vs. immediate clamping [59, 61, 64, 75–77, 82] and three studies comparing milking vs. immediate cord clamping [66, 69, 71]. The intervention was associated with lower mortality rates (5% vs. 7%) and the confidence interval included a potential beneficial effect (RR 0.70, 95% CI 0.51, 0.96; 10 RCTs, participants = 2,476, low certainty of evidence).

*1.2.1 Delayed vs. immediate clamping*

Five studies reported on neonatal mortality during initial hospitalization in the delayed vs. immediate clamping subgroup [59, 60, 64, 65, 80]. We are uncertain of the effect of delayed cord clamping compared to immediate clamping on the rates of all-cause mortality during initial hospitalization, (RR 0.51, 95% CI 0.26 to 1.00; participants = 595; studies = 5). This suggests that if the all-cause mortality during initial hospitalization following immediate clamping is 12%, the rate of all-cause mortality during initial hospitalization following delayed cord clamping would be between 3% and 12%.

*1.2.2 Milking vs. immediate clamping*

Two studies reported on neonatal mortality during initial hospitalization in the milking vs. immediate clamping subgroup [70, 74]. We are uncertain of the effect of milking compared to immediate clamping on the rates of all-cause mortality during initial hospitalization (RR 0.46, 95% CI 0.11 to 2.00; participants = 139; studies = 2; $I^2$ = 0%). This suggests that if the all-cause mortality during initial hospitalization following immediate clamping is 14%, the rate of all-cause mortality during initial hospitalization following delayed cord clamping would be between 2% and 28%.

*1.3 Major neurodevelopmental disability in children alive at 24 months of age.* One study compared delayed vs. immediate clamping and reported on major neurodevelopmental disability in children alive at 24 months on age [64].

*1.3.1 Motor disability*

One study reported on motor disability in children alive at 24 months of age; comparing delayed vs. immediate clamping [64]. We are uncertain of the effect of delayed compared to

immediate clamping on the rates of motor disability in children alive at 24 months of age (RR 0.41, 95% CI 0.08 to 2.06; participants = 276; studies = 1; $I^2$ = 0% very low certainty of evidence). This suggests that if the all motor disability following immediate clamping is 4%, the rate of motor disability following delayed cord clamping would be between 0% and 8%.

### 1.3.2. Cognitive disability

One study reported on cognitive disability in children alive at 24 months of age; comparing delayed vs. immediate clamping [64]. We are uncertain of the effect of delayed compared to immediate clamping on the rates of cognitive disability in children alive at 24 months of age (RR 1.86, 95% CI 0.71 to 4.89; participants = 276; studies = 1; $I^2$ = 0%). This suggests that if the all cognitive disability following immediate clamping is 4%, the rate of cognitive disability following delayed cord clamping would be between 3% and 20%.

### 1.3.3. Hearing disability

One study reported on hearing disability in children alive at 24 months of age; comparing delayed vs. immediate clamping [64]. We are uncertain of the effect of delayed compared to immediate clamping on the rates of hearing disability in children alive at 24 months of age (RR 1.01, 95% CI 0.14 to 7.10; participants = 276; studies = 1; $I^2$ = 0%). This suggests that if the all hearing disability following immediate clamping is 1%, the rate of hearing disability following delayed cord clamping would be between 0% and 7%.

### 1.3.4 Speech/language disability

One study reported on speech/language disability in children alive at 24 months of age; comparing delayed vs. immediate clamping [64]. We are uncertain of the effect of delayed compared to immediate clamping on the rates of speech/language disability in children alive at 24 months of age (RR 0.65, 95% CI 0.29 to 1.46; participants = 276; studies = 1; $I^2$ = 0%). This suggests that if the all speech/language disability following immediate clamping is 10%, the rate of speech/language disability following delayed cord clamping would be between 3% and 15%.

### 1.3.5 Vision disability

One study reported on vision disability in children alive at 24 months of age; comparing delayed vs. immediate clamping [64]. We are uncertain of the effect of delayed compared to immediate clamping on the rates of vision disability in children alive at 24 months of age (RR 1.01, 95% CI 0.14 to 7.10; participants = 276; studies = 1; $I^2$ = 0%). This suggests that if the all vision disability following immediate clamping is 1%, the rate of vision disability following delayed cord clamping would be between 0% and 7%

**1.4 Major neurodevelopmental disability in children alive at 3 years of age.** One study reported on major neurodevelopmental disability in children alive at 3 years of age; comparing milking vs. immediate clamping [66].

### 1.4.1 Cognitive disability

One study reported on cognitive disability in children alive at 3 years of age; comparing delayed vs. immediate clamping [66]. We are uncertain of the effect of delayed compared to immediate clamping on the rates of cognitive disability in children alive at 3 years of age (RR 0.14, 95% CI 0.01 to 2.6; participants = 73; studies = 1; $I^2$ = 0% very low certainty of evidence). This suggests that if the all cognitive disability following immediate clamping is 8%, the rate of cognitive disability following delayed cord clamping would be between 0% and 21%.

### 1.4.2 Vision disability

One study reported on vision disability in children alive at 3 years of age; comparing delayed vs. immediate clamping [66]. The study authors did not report any case of vision disability in the study.

### 1.4.3 Hearing disability

One study reported on hearing disability in children alive at 3 years of age; comparing delayed vs. immediate clamping [66]. We are uncertain of the effect of delayed compared to

immediate clamping on the rates of hearing disability in children alive at 3 years of age; (RR 0.32, 95% CI 0.01 to 7.71; participants = 73; studies = 1; $I^2$ = 0% $I^2$ = 0%). This suggests that if the all hearing disability following immediate clamping is 3%, the rate of hearing disability following delayed cord clamping would be between 0% and 23%.

*1.4.4 Cerebral palsy*

One study reported on cerebral palsy in children alive at 3 years of age; comparing delayed vs. immediate clamping [66]. We are uncertain of the effect of delayed compared to immediate clamping on the rates of cerebral palsy in children alive at 3 years of age; (RR 5.84, 95% CI 0.74 to 46.11; participants = 73; studies = 1; $I^2$ = 0%). This suggests that if the all cerebral palsy following immediate clamping is 3%, the rate of cerebral palsy following delayed cord clamping would be between 2% and 138%.

*1.5 Death or adverse neurodevelopmental outcome for children 18 to 24 months*. One study reported on death or adverse neurodevelopmental outcome for children 18 to 24 months comparing delayed vs. immediate clamping [64]. The intervention was associated with lower rates of mortality and adverse neurodevelopmental outcome (12% vs. 17%) and the confidence interval included a potential beneficial effect (RR 0.70, 95% CI 0.44 to 1.11; participants = 276; studies = 1; $I^2$ = 0% very low certainty of evidence)

*1.6 ROP—any*. Nine studies reported on any ROP; seven studies comparing delayed vs. immediate clamping [59, 63, 67, 68, 74, 75, 79] and two studies comparing milking vs. immediate cord clamping [66, 67]. We are uncertain of the effect of delayed cord clamping or milking compared to immediate clamping or no milking on the rates of ROP of any grade RR 0.92, 95% CI 0.72 to 1.19; participants = 520; studies = 9; $I^2$ = 0%). This suggests that if the rate of any grade ROP following immediate clamping or no milking is 29%, the rate of any grade ROP following delayed cord clamping or milking compared would be between 21% and 35%.

*1.6.1 Delayed vs. immediate clamping*

Seven studies reported on any ROP in the delayed vs. immediate clamping [59, 63, 67, 68, 74, 75, 79] subgroup. We are uncertain of the effect of delayed cord clamping compared to immediate clamping the rates of ROP of any grade (RR 0.93, 95% CI 0.72 to 1.20; participants = 416; studies = 7; $I^2$ = 0%). This suggests that if the rate of any grade ROP following immediate clamping 34%, the rate of any grade ROP following delayed cord clamping would be between 24% and 41%.

*1.6.2 Milking vs. immediate clamping*

Two studies reported on any ROP in the milking vs. immediate clamping [66, 67] subgroup. We are uncertain of the effect of milking compared to immediate clamping the rates of ROP of any grade (RR 0.65, 95% CI 0.12 to 3.66; participants = 104; studies = 2; $I^2$ = 100%). This suggests that if the rate of any grade ROP following immediate clamping is 6%, the rate of any grade ROP following delayed cord clamping would be between 1% and 22%.

*1.7 ROP—severe*. Four studies reported on severe ROP; three studies comparing delayed vs. immediate clamping [62, 64, 82] and one study comparing milking vs. immediate cord clamping [58]. We are uncertain of the effect of delayed cord clamping or milking compared to immediate clamping on the rates of ROP of severe grade (RR 0.77, 95% CI 0.53 to 1.13; participants = 2011; studies = 4; $I^2$ = 0%). This suggests that if the rate of severe grade ROP following immediate clamping is 6%, the rate of severe grade ROP following delayed cord clamping or milking compared would be between 3% and 7%.

*1.7.1 Delayed vs. immediate clamping*

Three studies reported on severe ROP in the delayed vs. immediate clamping subgroup [62, 64, 82]. We are uncertain of the effect of delayed cord clamping compared to immediate clamping on the rates of ROP of severe grade (RR 0.78, 95% CI 0.53 to 1.15; participants = 1963; studies = 3; $I^2$ = 0%). This suggests that if the rate of severe grade ROP following immediate

clamping is 6%, the rate of severe grade ROP following delayed cord clamping compared would be between 3% and 7%.

*1.7.2 Milking vs. immediate clamping*

One study reported on severe ROP in the milking vs. immediate clamping subgroup [58]. We are uncertain of the effect of milking compared to immediate clamping on the rates of ROP of severe grade (RR 0.5, 95% CI 0.05 to 5.15; participants = 48; studies = 1; $I^2$ = 0%). This suggests that if the rate of severe grade ROP following immediate clamping is 8%, the rate of severe grade ROP following delayed cord clamping compared would be between 0% and 41%.

*1.8 IVH—any.* Eighteen studies reported on any IVH; fourteen studies comparing delayed vs. immediate clamping [59, 61–65, 68, 72, 74–77, 79, 80] and four studies comparing milking vs. immediate cord clamping [66, 70, 71, 81]. We are uncertain of the effect of delayed cord clamping or milking compared to immediate clamping or no milking on the rates of IVH of any grade (RR 0.88, 95% CI 0.73 to 1.06; participants = 1544; studies = 18; $I^2$ = 19%). This suggests that if the rate of any grade IVH following immediate clamping or no milking is 24%, the rate of any grade IVH following delayed cord clamping or milking compared would be between 18% and 25%.

*1.8.1 Delayed vs. immediate clamping*

Fourteen studies reported on any IVH in the delayed vs. immediate clamping subgroup [59, 61–65, 68, 72, 74–77, 79, 80]. We are uncertain of the effect of delayed cord clamping compared to immediate clamping on the rates of IVH of any grade (RR 0.86, 95% CI 0.70 to 1.05; participants = 1310; studies = 14; $I^2$ = 34%). This suggests that if the rate of any grade IVH following immediate clamping is 25%, the rate of any grade IVH following delayed cord clamping compared would be between 18% and 26%.

*1.8.2 Milking vs. immediate clamping*

Four studies reported on any IVH in the milking vs. immediate cord clamping subgroup [66, 70, 71, 81]. We are uncertain of the effect of milking compared to immediate clamping on the rates of IVH of any grade (RR 0.99, 95% CI 0.64 to 1.53; participants = 234; studies = 4; $I^2$ = 0%). This suggests that if the rate of any grade IVH following immediate clamping is 24%, the rate of any grade IVH following milking compared would be between 15% and 37%.

*1.9 IVH—severe.* Twelve studies reported on severe IVH; eight studies comparing delayed vs. immediate clamping [59, 61, 64, 65, 67, 76, 80, 82] and four studies comparing milking vs. immediate cord clamping [58, 66, 67, 71]. We are uncertain of the effect of delayed cord clamping or milking compared to immediate clamping or no milking on the rates of IVH of any grade (RR 0.97, 95% CI 0.65 to 1.43; participants = 2535; studies = 12; $I^2$ = 0%). This suggests that if the rate of any grade IVH following immediate clamping or no milking is 4%, the rate of any grade IVH following delayed cord clamping or milking compared would be between 3% and 6%.

*1.9.1 Delayed vs. immediate clamping*

Eight studies reported on severe IVH in the delayed vs. immediate clamping subgroup [59, 61, 64, 65, 67, 76, 80, 82]. We are uncertain of the effect of delayed cord clamping compared to immediate clamping on the rates of IVH of any grade (RR 0.98, 95% CI 0.62 to 1.54; participants = 2324; studies = 8; $I^2$ = 0%). This suggests that if the rate of any grade IVH following immediate clamping is 3%, the rate of any grade IVH following delayed cord clamping compared would be between 2% and 4%.

*1.9.2 Milking vs. immediate clamping*

Four studies reported on severe IVH in the milking vs. immediate cord clamping subgroup [58, 66, 67, 71]. We are uncertain of the effect of milking compared to immediate clamping on the rates of IVH of any grade (RR 0.94, 95% CI 0.44 to 2.02; participants = 211; studies = 4; $I^2$ = 6%). This suggests that if the rate of any grade IVH following immediate clamping is 11%, the rate of any grade IVH following milking compared would be between 5% and 22%.

*1.10 White matter at term-equivalent/ MRI abnormalities at term equivalent.* Three studies reported on white matter at term-equivalent/MRI abnormalities at term equivalent comparing delayed vs. immediate clamping [65, 72, 82]. We are uncertain of the effect of delayed cord clamping compared to immediate clamping on the rates of white matter at term-equivalent/ MRI abnormalities at term equivalent (RR 1.12, 95% CI 0.76 to 1.64; participants = 1904; studies = 3; $I^2$ = 0%). This suggests that if the rate of white matter at term-equivalent/MRI abnormalities at term equivalent following immediate clamping is 5%, the rate of white matter at term-equivalent/MRI abnormalities at term equivalent following delayed cord clamping compared would be between 4% and 8%

*1.11 BPD chronic lung disease.* Twelve studies reported on BPD/chronic lung disease; eight studies comparing delayed vs. immediate clamping [59, 63–65, 67, 76, 77, 82] and four studies comparing milking vs. immediate cord clamping [58, 66, 67, 71]. We are uncertain of the effect of delayed cord clamping or milking compared to immediate clamping or no milking on the rates of BPD/chronic lung disease (RR 1.08, 95% CI 0.99 to 1.18; participants = 2760; studies = 12; $I^2$ = 0%). This suggests that if the rate of BPD/chronic lung disease following immediate clamping is 39%, the rate of BPD/chronic lung disease following delayed cord clamping or milking compared would be between 39% and 46%.

*1.11.1 Delayed vs. immediate clamping*

Eight studies reported on BPD/chronic lung disease in the delayed vs. immediate clamping subgroup [59, 63–65, 67, 76, 77, 82]. We are uncertain of the effect of delayed cord clamping compared to immediate clamping on the rates of BPD/chronic lung disease (RR 1.10, 95% CI 1.00 to 1.20; participants = 2549; studies = 8; $I^2$ = 0%). This suggests that if the rate of BPD/ chronic lung disease following immediate clamping is 39%, the rate of BPD/chronic lung disease following delayed cord clamping compared would be between 39% and 47%.

*1.11.2 Milking vs. immediate clamping*

Four studies reported on BPD/chronic lung disease in the milking vs. immediate cord clamping subgroup [58, 66, 67, 71]. We are uncertain of the effect of milking compared to immediate clamping on the rates of BPD/chronic lung disease (RR 0.83, 95% CI 0.55 to 1.23; participants = 211; studies = 4; $I^2$ = 31%). This suggests that if the rate of BPD/chronic lung disease following immediate clamping is 36%, the rate of BPD/chronic lung disease following milking compared would be between 20% and 44%.

*1.12 NEC.* Eighteen studies reported on NEC; fourteen studies comparing delayed vs. immediate clamping [59, 63–65, 67, 68, 73–77, 79, 80, 82] and four studies comparing milking vs. immediate cord clamping [58, 66, 67, 70]. We are uncertain of the effect of delayed cord clamping or milking compared to immediate clamping or no milking on the rates of NEC (RR 0.80, 95% CI 0.62 to 1.02; participants = 3022; studies = 18; $I^2$ = 0). This suggests that if the rate of NEC following immediate clamping is 7%, the rate of NEC following delayed cord clamping or milking compared would be between 4% and 7%.

*1.12.1. Delayed vs. immediate clamping*

Fourteen studies reported on NEC in the delayed vs. immediate clamping subgroup [59, 63–65, 67, 68, 73–77, 79, 80, 82]. We are uncertain of the effect of delayed cord clamping compared to immediate clamping on the rates of NEC (RR 0.79, 95% CI 0.61 to 1.01; participants = 2845; studies = 14; $I^2$ = 0%). This suggests that if the rate of NEC following immediate clamping is 7%, the rate of NEC following delayed cord clamping compared would be between 4% and 7%.

*1.12.2 Milking vs. immediate clamping*

Four studies reported on NEC in the milking vs. immediate cord clamping subgroup [58, 66, 67, 70]. We are uncertain of the effect of milking compared to immediate clamping on the rates of NEC (RR 1.06, 95% CI 0.37 to 3.03; participants = 177; studies = 4; $I^2$ = 0%). This

suggests that if the rate of NEC following immediate clamping is 7%, the rate of NEC following milking compared would be between 3% and 21%.

*1.13 Volume in ml of blood withdrawn until hospital discharge*. Three studies reported on volume in ml of blood withdrawn until hospital discharge comparing delayed vs. immediate clamping [59, 76, 79]. We are uncertain of the effect of delayed cord clamping compared to immediate clamping on the rates of volume in ml of blood withdrawn until hospital discharge (MD -0.20, 95% CI -2.76 to 2.35; participants = 145; studies = 3; $I^2$ = 0%).

*1.14 Volume in ml of blood transfused until hospital discharge*. Two studies reported on volume in ml of blood transfused until hospital discharge comparing delayed vs. immediate clamping [75, 76]. We are uncertain of the effect of delayed cord clamping compared to immediate clamping on the rates of volume in ml of blood transfused until hospital discharge (MD -10.70, 95% CI -27.60 to 6.20; participants = 104; studies = 2; $I^2$ = 0%).

*1.15 Number of blood transfusions until hospital discharge*. Four studies reported on the number of blood transfusions until hospital discharge; three studies comparing delayed vs. immediate clamping [59, 68, 73] and one study comparing milking vs. immediate cord clamping [70]. We are uncertain of the effect of delayed cord clamping or milking compared to immediate clamping or no milking on the rates of number of blood transfusions until hospital discharge (MD -0.05, 95% CI -0.44 to 0.34; participants = 144; studies = 4; $I^2$ = 0%).

*1.15.1 Delayed vs. immediate cord clamping*

Three studies reported on the number of blood transfusions until hospital discharge in the delayed vs. immediate clamping subgroup [59, 68, 73]. We are uncertain of the effect of delayed cord clamping compared to immediate clamping on the rates of number of blood transfusions until hospital discharge (MD -0.06, 95% CI -0.47 to 0.35; participants = 118; studies = 3; $I^2$ = 0%).

*1.15.2 Milking vs. immediate clamping*

One study reported on number of blood transfusions until hospital discharge in the milking vs. immediate cord clamping subgroup [70]. We are uncertain of the effect of milking compared to immediate clamping or no milking on the rates of number of blood transfusions until hospital discharge (MD 0.10, 95% CI -1.25 to 1.45; participants = 26; studies = 1; $I^2$ = 0%).

*1.16 Need for blood transfusions until hospital discharge*. Nine studies reported on need for blood transfusions until hospital discharge; seven studies comparing delayed vs. immediate clamping [61–65, 76, 80] and two studies comparing milking vs. immediate cord clamping. The intervention was associated with lower need for blood transfusions until hospital discharge (35% vs. 44%) and the confidence interval included a potential beneficial effect (RR 0.82, 95% CI 0.70 to 0.96; participants = 877; studies = 9; $I^2$ = 54%).

*1.16.1 Delayed vs. immediate clamping*

Seven studies reported on the need for blood transfusions until hospital discharge in the delayed vs. immediate clamping subgroup [61–65, 76, 80]. The intervention was associated with lower need for blood transfusions until hospital discharge (32% vs. 41%) and the confidence interval included a potential beneficial effect (RR 0.82, 95% CI 0.69 to 0.98; participants = 769; studies = 7; $I^2$ = 47%).

*1.16.2 Milking vs. immediate clamping*

Two studies reported on the need for blood transfusions until hospital discharge in the milking vs. immediate cord clamping subgroup [58, 71]. We are uncertain of the effect of milking compared to immediate clamping on the rates of need for blood transfusions until hospital discharge (RR 0.80, 95% CI 0.56 to 1.15; participants = 108; studies = 2; $I^2$ = 84%). This suggests that if the rate of need for blood transfusions until hospital discharge following immediate clamping is 48%, the rate of need for blood transfusions until hospital discharge following milking compared would be between 27% and 55%.

*1.17 Concentrations of total hemoglobin (Hb)—day 1–7.* Seventeen studies reported on concentrations of total hemoglobin (Hb) from day one to seven; thirteen studies comparing delayed vs. immediate clamping [59–63, 65, 68, 73, 75–79] and four studies comparing milking vs. immediate cord clamping [66, 70, 71, 81].

*1.17.1 Delayed vs. immediate clamping*

Thirteen studies reported on concentrations of total hemoglobin (Hb) from day one to seven in the delayed vs. immediate clamping subgroup [59–63, 65, 68, 73, 75, 76, 78, 79]. The concentration of total hemoglobin (Hb) from day one to seven was higher in the DCC group (MD 2.39, 95% CI 2.17 to 2.61; participants = 921; studies = 13; $I^2$ = 92%).

*1.17.2 Milking vs. immediate clamping*

Four studies reported on concentrations of total hemoglobin (Hb) from day one to seven in the milking vs. immediate cord clamping subgroup [66, 70, 71, 81]. The concentrations of total hemoglobin (Hb) from day one to seven was higher in the milking group (MD 0.97, 95% CI 0.37 to 1.58; participants = 234; studies = 4; $I^2$ = 0%)

*1.18 Concentrations of total hemoglobin (Hb)—day 8–14.* Two studies reported on concentrations of total hemoglobin (Hb) from day eight to fourteen, comparing delayed vs. immediate clamping [59, 79]. We are uncertain of the effect of delayed compared to immediate clamping on the concentrations of total hemoglobin (Hb) from day eight to fourteen (MD 0.62, 95% CI -0.16 to 1.40; participants = 73; studies = 2; $I^2$ = 0%)

*1.19 Concentrations of total hemoglobin (Hb)—day 15+.* Two studies reported on concentrations of total hemoglobin (Hb) after day fifteen, comparing delayed vs. immediate clamping [59, 79]. We are uncertain of the effect of delayed compared to immediate clamping on the concentrations of total hemoglobin (Hb) after day fifteen (MD 0.30, 95% CI -0.54 to 1.14; participants = 73; studies = 2; $I^2$ = 0%)

*1.20 Late sepsis until hospital discharge.* Sixteen studies reported on late sepsis until discharge; thirteen studies comparing delayed vs. immediate clamping [59–64, 67, 68, 73, 74, 76, 79, 82] and three studies comparing milking vs. immediate cord clamping [66, 67, 81]. We are uncertain of the effect of delayed cord clamping or milking compared to immediate clamping or no milking on the rates of late sepsis until discharge (RR 0.98, 95% CI 0.86 to 1.12; participants = 2670; studies = 16; $I^2$ = 10%). This suggests that if the rate of late sepsis until discharge following immediate clamping is 22%, the rate of late sepsis until discharge following delayed cord clamping or milking compared would be between 19% and 25%.

*1.20.1 Delayed vs. immediate clamping*

Thirteen studies reported on late sepsis until discharge in the delayed vs. immediate clamping subgroup [59–64, 67, 68, 73, 74, 76, 79, 82]. We are uncertain of the effect of delayed cord clamping compared to immediate clamping on the rates of late sepsis until discharge (RR 0.97, 95% CI 0.85 to 1.10; participants = 2354; studies = 9; $I^2$ = 0%). This suggests that if the rate of late sepsis until discharge following immediate clamping is 22%, the rate of late sepsis until discharge following delayed cord clamping compared would be between 19% and 24%.

*1.20.2 Milking vs. immediate clamping*

Three studies reported on late sepsis until discharge in the milking vs. immediate cord clamping subgroup [66, 67, 81]. We are uncertain of the effect of milking compared to immediate clamping on the rates of late sepsis until discharge (RR 1.33, 95% CI 0.66 to 2.68; participants = 178; studies = 3; $I^2$ = 0%). This suggests that if the rate of late sepsis until discharge following immediate clamping is 13%, the rate of late sepsis until discharge following milking compared would be between 9% and 35%.

*1.21 PDA (pharmacological treatment and surgical treatment).* Thirteen studies reported on PDA; nine studies comparing delayed vs. immediate clamping [17, 59, 62, 64, 65, 68, 73, 79, 82] and four studies comparing milking vs. immediate cord clamping [58, 66, 71, 81]. We are

uncertain of the effect of delayed cord clamping or milking compared to immediate clamping or no milking on the rates of PDA (RR 0.98, 95% CI 0.87 to 1.10; participants = 2610; studies = 13; $I^2$ = 0%). This suggests that if the rate of PDA following immediate clamping is 28%, the rate of PDA following delayed cord clamping or milking compared would be between 24% and 31%.

*1.21.1 Delayed vs. immediate clamping*

Nine studies reported on PDA (pharmacological treatment and surgical treatment) in the delayed vs. immediate clamping subgroup [59, 62, 64, 65, 68, 73, 79, 80, 82]. We are uncertain of the effect of delayed cord clamping compared to immediate clamping on the rates of PDA (pharmacological treatment and surgical treatment) (RR 0.97, 95% CI 0.85 to 1.10; participants = 2354; studies = 9; $I^2$ = 0%). This suggests that if the rate of PDA (pharmacological treatment and surgical treatment) following immediate clamping is 28%, the rate of PDA (pharmacological treatment and surgical treatment) following delayed cord clamping compared would be between 24% and 31%.

*1.21.2 Milking vs. immediate clamping*

Four studies reported on PDA (pharmacological treatment and surgical treatment) in the milking vs. immediate cord clamping subgroup [58, 66, 71, 81]. We are uncertain of the effect of milking compared to immediate clamping on the rates of PDA (pharmacological treatment and surgical treatment) (RR 1.05, 95% CI 0.71 to 1.54; participants = 256; studies = 4; $I^2$ = 0%). This suggests that if the rate of PDA (pharmacological treatment and surgical treatment) following immediate clamping is 27%, the rate of PDA (pharmacological treatment and surgical treatment) following milking compared would be between 19% and 42%.

*1.22 Duration in days of respiratory support (CPAP).* One study reported on the duration in days of respiratory support (CPAP) [73], comparing delayed vs. immediate clamping. We are uncertain of the effect of delayed cord clamping compared to immediate clamping on the rates of the duration in days of respiratory support (CPAP) (MD 4.25, 95% CI -5.86 to 14.36; participants = 36; studies = 1; $I^2$ = 0%)

*1.23 Duration in days of respiratory support (ventilator).* Two studies reported on the duration in days of respiratory support (ventilator) [68, 73], comparing delayed vs. immediate clamping. We are uncertain of the effect of delayed compared to immediate clamping on the duration in days of respiratory support (ventilator) (MD -0.34, 95% CI -2.69 to 2.02; participants = 78; studies = 2; $I^2$ = 0%)

*1.24 Duration in days of supplemental oxygen requirement.* Three studies reported on the duration in days of supplemental oxygen requirement [68, 73, 75], comparing delayed vs. immediate clamping. We are uncertain of the effect of delayed compared to immediate clamping on the duration in days of supplemental oxygen requirement (MD -5.01, 95% CI -14.52 to 4.51; participants = 110; studies = 3; $I^2$ = 0%)

*1.25 Volume in ml/kg of blood transfused until hospital discharge.* Two studies reported on the volume in ml/kg of blood transfused until hospital discharge [63, 79], comparing delayed vs. immediate clamping. The results indicated a potential reduction in volume in ml/kg of blood transfused until hospital discharge in the delayed cord clamping group compared to the immediate clamping (MD -5.20, 95% CI -5.45 to -4.95; participants = 123; studies = 2; $I^2$ = 0%)

*1.26 Impaired motor skills.* Three studies reported on impaired motor skills; two studies comparing delayed vs. immediate clamping [64, 77] and one study comparing milking vs. immediate cord clamping [66]. We are uncertain of the effect of delayed cord clamping or milking compared to immediate clamping on motor skills (RR 0.72, 95% CI 0.44 to 1.17; participants = 459; studies = 2; $I^2$ = 0%; RR 0.97, 95% CI 0.41 to 2.31; participants = 73; studies = 1; $I^2$ = 100%)

No data were reported for the following outcomes: one-year survival, concentration of fetal hemoglobin (HbF), duration in days of hospital stay, poor academic performance at 12 years of age, behavioral problem, time in minutes to perform the procedure, pain during device

insertion/use and blood sampling, number of skin-breaking procedures associated to blood testing, insertion and repositioning of the device, skin/soft tissue injury associated to blood testing, insertion and repositioning of the device, site infection associated to blood testing, insertion and repositioning of the device, and thrombotic event rates.

## Comparison 2: Blood sampling from the umbilical cord or from the placenta vs. blood sampling from the infant

Two studies with 124 infants compared blood sampling from the umbilical cord or from the placenta vs. blood sampling from the infant [83, 84].

**2.1 All-cause mortality during initial hospitalization.** One study reported on all-cause mortality during initial hospitalization [83]. We are uncertain of the effect of blood sampling from the umbilical cord or from the placenta compared to blood sampling from the infant on the all-cause mortality rate (RR 1.00, 95% CI 0.15 to 6.76; one study, 80 infants; very low certainty evidence). This suggests that if the rate of all-cause mortality following blood sampling from the infant is 5%, the rate of all-cause mortality following blood sampling from the umbilical cord or from the placenta would be between 1% and 34%.

**2.2 ROP–any.** One study reported on any ROP [83]. We are uncertain of the effect of blood sampling from the umbilical cord or from the placenta compared to blood sampling from the infant on the rate of any ROP (RR 0.60, 95% CI 0.34 to 1.06; participants = 80; studies = 1; $I^2$ = 0%). This suggests that if the rate of any ROP following blood sampling from the infant is 50%, the rate of any ROP following blood sampling from the umbilical cord or from the placenta would be between 17% and 53%.

**2.3 ROP–severe.** One study reported on severe ROP [84]. We are uncertain of the effect of blood sampling from the umbilical cord or from the placenta compared to blood sampling from the infant on the rate of severe ROP (RR 0.56, 95% CI 0.10 to 3.00; participants = 44; studies = 1; $I^2$ = 0%). This suggests that if the rate of severe ROP following blood sampling from the infant is 15%, the rate of severe ROP following blood sampling from the umbilical cord or from the placenta would be between 2% and 45%.

**2.4 IVH–severe.** Two studies reported on severe IVH [83, 84]. We are uncertain of the effect of blood sampling from the umbilical cord or from the placenta compared to blood sampling from the infant on the rate of severe IVH (RR 0.43, 95% CI 0.14 to 1.31; participants = 124; studies = 2; $I^2$ = 0%). This suggests that if the rate of severe IVH following blood sampling from the infant is 13%, the rate of severe IVH following blood sampling from the umbilical cord or from the placenta would be between 2% and 17%.

**2.5 BPD/chronic lung disease.** One study reported on BPD [83]. We are uncertain of the effect of blood sampling from the umbilical cord or from the placenta compared to blood sampling from the infant on the rate of BPD (RR 0.88, 95% CI 0.51 to 1.51; participants = 80; studies = 1; $I^2$ = 0%). This suggests that if the rate of BPD following blood sampling from the infant is 43%, the rate of BPD following blood sampling from the umbilical cord or from the placenta would be between 22% and 65%.

**2.6 NEC.** One study reported on necrotizing enterocolitis [83]. We are uncertain of the effect of blood sampling from the umbilical cord or from the placenta compared to blood sampling from the infant on the rate of necrotizing enterocolitis (RR 0.50, 95% CI 0.05 to 5.30; participants = 80; studies = 1; $I^2$ = 0%). This suggests that if the rate of necrotizing enterocolitis following blood sampling from the infant is 5%, the rate of necrotizing enterocolitis following blood sampling from the umbilical cord or from the placenta would be between 0% and 27%.

**2.7 Volume in mL of blood transfused until hospital discharge.** One study reported on volume in ml of blood transfused until hospital discharge [84]. We are uncertain of the effect

of blood sampling from the umbilical cord or from the placenta compared to blood sampling from the infant on the volume in ml of blood transfused until hospital discharge (MD -29.26, 95% CI -36.88 to -21.64; participants = 44; studies = 1; $I^2$ = 0%).

**2.8 Number of blood transfusions until hospital discharge.** One study reported on the number of blood transfusions until hospital discharge [84]. We are uncertain of the effect of blood sampling from the umbilical cord or from the placenta compared to blood sampling from the infant on the number of blood transfusions until hospital discharge (MD -2.09, 95% CI -2.91 to -1.27; participants = 44; studies = 1; $I^2$ = 0%).

**2.9 Need for blood transfusions until hospital discharge.** One study reported on the need for blood transfusions until hospital discharge [83]. We are uncertain of the effect of blood sampling from the umbilical cord or from the placenta compared to blood sampling from the infant on the need for blood transfusions until hospital discharge (RR 0.91, 95% CI 0.75 to 1.11; participants = 80; studies = 1; $I^2$ = 0%). This suggests that if the need for blood transfusions until hospital discharge following blood sampling from the infant is 88%, the need for blood transfusions until hospital discharge following blood sampling from the umbilical cord or from the placenta would be between 66% and 98%.

**2.10 Concentration of total hemoglobin (Hb)—day 1–7.** Two studies reported on the concentration of total hemoglobin (Hb) from day one to seven [83, 84]. We are uncertain of the effect of blood sampling from the umbilical cord or from the placenta compared to blood sampling from the infant on the concentration of total hemoglobin (Hb) from day one to seven (MD -0.38, 95% CI -1.02 to 0.26; participants = 124; studies = 2; $I^2$ = 0%).

**2.11 Concentration of total hemoglobin (Hb)—day 15+.** One study reported on the concentration of total hemoglobin (Hb) after day 15 [83]. We are uncertain of the effect of blood sampling from the umbilical cord or from the placenta compared to blood sampling from the infant on the concentration of total hemoglobin (Hb) after day 15 (MD -0.20, 95% CI -0.81 to 0.41; participants = 80; studies = 1; $I^2$ = 0%).

**2.12 Late sepsis until hospital discharge.** One study reported on late sepsis until hospital discharge [83]. We are uncertain of the effect of blood sampling from the umbilical cord or from the placenta compared to blood sampling from the infant on the rate of late sepsis until hospital discharge (RR 1.00, 95% CI 0.47 to 2.14; participants = 80; studies = 1; $I^2$ = 0%). This suggests that if the rate of late sepsis until hospital discharge following blood sampling from the infant is 25%, the rate of late sepsis until hospital discharge following blood sampling from the umbilical cord or from the placenta would be between 12% and 54%.

**2.13 PDA (pharmacological treatment and surgical treatment).** One study reported on PDA [83]. We are uncertain of the effect of blood sampling from the umbilical cord or from the placenta compared to blood sampling from the infant on the rate of PDA (RR 1.09, 95% CI 0.76 to 1.56; participants = 80; studies = 1; $I^2$ = 100%). This suggests that if the rate of PDA following blood sampling from the infant is 58%, the rate of PDA following blood sampling from the umbilical cord or from the placenta would be between 44% and 90%.

**2.14 Duration in days of respiratory support.** One study reported on the duration in days of respiratory support [83]. We are uncertain of the effect of blood sampling from the umbilical cord or from the placenta compared to blood sampling from the infant on the duration in days of respiratory support (presented in median and interquartile range: 31 days (16 to 56 days) in intervention group vs. 34 days (15 to 59 days) in control group (P = 0.98; one study, 69 infants).

**2.15 Duration in days of hospital stay.** One study reported on the duration in days of hospital stay [83]. We are uncertain of the effect of blood sampling from the umbilical cord or from the placenta compared to blood sampling from the infant on the duration in days of hospital stay (presented in median and interquartile range: 64 days (50 to 85 days) in intervention group vs. 70 days (54 to 93 days) in control group (P = 0.24; one study, 80 infants).

No data were reported for the following outcomes: all-cause neonatal mortality (first 28 days of life), one-year survival, major neurodevelopmental disability *for children 18 to 24 months*, major neurodevelopmental disability *for children 3 to 5 years of age*, mortality or major neurodevelopmental disability [composite outcome], any germinal matrix-IVH, white matter at term-equivalent MRI abnormalities at term equivalent age, concentration of total hemoglobin (Hb) day 8–14, concentration of fetal hemoglobin (HbF), duration in days of supplemental oxygen requirement, each component of the composite outcome 'major neurodevelopmental disability,' poor academic performance at 12 years of age, motor skills problem, behavioral problem, time in minutes to perform the procedure, pain during device insertion/use and blood sampling, number of skin-breaking procedures associated to blood testing, insertion and repositioning of the device, skin/soft tissue injury associated to blood testing, insertion and repositioning of the device, site infection associated to blood testing, insertion and repositioning of the device, and thrombotic event rates.

## Comparison 3: Devices to reintroduce the blood after analysis vs. conventional blood sampling

One study with 93 infants compared devices to reintroduce blood after analysis vs. conventional blood sampling [24].

**3.1 All-cause mortality during initial hospitalization.** One study reported on all-cause mortality during initial hospitalization [24]. We are uncertain of the effect of devices to reintroduce blood after analysis compared to conventional blood sampling on the all-cause mortality rate (RR 1.53, 95% CI 0.59 to 3.96; participants = 93; studies = 1; $I^2$ = 0%, very low certainty evidence). This suggests that if the rate of all-cause mortality following conventional blood sampling is 13%, the rate of all-cause mortality following the use of devices to reintroduce blood after analysis would be between 8% and 51%.

**3.2 Major neurodevelopmental disability.** One study reported on major neurodevelopmental disability [24]. We are uncertain of the effect of devices to reintroduce blood after analysis compared to conventional blood sampling on major neurodevelopmental disability (MD -1.00, 95% CI -2.90 to 0.90; one study, 48 infants; very low certainty of evidence).

**3.3 ROP–any.** One study reported on any ROP [24]. We are uncertain of the effect of devices to reintroduce blood after analysis compared to conventional blood sampling on any ROP (RR 1.36, 95% CI 0.32 to 5.75; participants = 93; studies = 1; $I^2$ = 0%). This suggests that if the rate of any ROP following conventional blood sampling is 6%, the rate of any ROP following the use of devices to reintroduce blood after analysis would be between 2% and 35%.

**3.4 IVH–severe.** One study reported on severe IVH [24]. We are uncertain of the effect of devices to reintroduce blood after analysis compared to conventional blood sampling on severe IVH (RR 1.17, 95% CI 0.46 to 2.96; participants = 93; studies = 1; $I^2$ = 0%). This suggests that if the rate of severe IVH following conventional blood sampling is 15%, the rate of severe IVH following the use of devices to reintroduce blood after analysis would be between 7% and 44%.

**3.5 NEC.** One study reported on NEC [24]. We are uncertain of the effect of devices to reintroduce blood after analysis compared to conventional blood sampling on the rate of NEC (RR 0.41, 95% CI 0.08 to 2.00; participants = 93; studies = 1; $I^2$ = 0%). This suggests that if the rate of NEC following conventional blood sampling is 11%, the rate of NEC following the use of devices to reintroduce blood after analysis would be between 1% and 22%.

**3.6 Concentration of total hemoglobin (Hb)—day 1–7.** One study reported on the concentration of total hemoglobin (Hb) from day one to seven [24]. We are uncertain of the effect of devices to reintroduce blood after analysis compared to conventional blood sampling on the

concentration of total hemoglobin (Hb) from day one to seven (MD 1.10, 95% CI -6.37 to 8.57; participants = 93; studies = 1; $I^2$ = 0%).

**3.7 Concentration of total hemoglobin (Hb)—day 8–14.** One study reported on the concentration of total hemoglobin (Hb) from day eight to 14 [24]. We are uncertain of the effect of devices to reintroduce blood after analysis compared to conventional blood sampling on the concentration of total hemoglobin (Hb) from day eight to 14 (MD -6.10, 95% CI -12.03 to -0.17; participants = 93; studies = 1; $I^2$ = 0%).

**3.8 PDA (pharmacological treatment and surgical treatment).** One study reported on PDA [24]. We are uncertain of the effect of devices to reintroduce blood after analysis compared to conventional blood sampling on the rate of PDA (RR 0.74, 95% CI 0.50 to 1.09; participants = 93; studies = 1; $I^2$ = 0%). This suggests that if the rate of PDA following conventional blood sampling is 62%, the rate of PDA following the use of devices to reintroduce blood after analysis would be between 31% and 68%.

**3.9 Duration in days of hospital stay.** One study reported on the duration in days of hospital stay [24]. We are uncertain of the effect of devices to reintroduce blood after analysis compared to conventional blood sampling on the duration in days of hospital stay (presented in median and interquartile range: 91 days (77 to 124 days) in intervention group vs. 102 days (86 to 121 days) in control group (P = 0.68; one study, 93 infants)).

No data were reported for the following outcomes: all-cause neonatal mortality (first 28 days of life), one-year survival, major neurodevelopmental disability *for children 18 to 24 months*, major neurodevelopmental disability *for children 3 to 5 years of age*, severe retinopathy of prematurity, any germinal matrix-IVH, white matter at term-equivalent MRI abnormalities at term equivalent age, BPD/chronic lung disease, volume in mL of blood withdrawn until hospital discharge, volume in mL of blood transfused until hospital discharge, number of blood transfusions until hospital discharge, need for blood transfusions until hospital discharge, concentration of total hemoglobin (Hb) day 15+, concentration of fetal hemoglobin (HbF), late sepsis until hospital discharge, duration in days of respiratory support, duration in days of supplemental oxygen requirement, each component of the composite outcome 'major neurodevelopmental disability,' poor academic performance at 12 years of age, motor skills problem, behavioral problem, time in minutes to perform the procedure, pain during device insertion/ use and blood sampling, number of skin-breaking procedures associated to blood testing, insertion and repositioning of the device, skin/soft tissue injury associated to blood testing, insertion and repositioning of the device, site infection associated to blood testing, insertion and repositioning of the device, and thrombotic event rates.

## Comparison 4: Devices to monitor glucose levels, subcutaneous vs. conventional blood sampling

Two studies with 98 infants compared devices to monitor glucose levels, subcutaneous vs. conventional blood sampling [85, 87].

**4.1 All-cause neonatal mortality (first 28 days).** One study reported on all-cause neonatal mortality (first 28 days) [85]. However, no events were observed.

**4.2 All-cause mortality during initial hospitalization.** One study reported on all-cause mortality during initial hospitalization [85]. We are uncertain of the effect of devices to monitor glucose levels, subcutaneous compared to conventional blood sampling on the all-cause mortality rate (RR 0.33, 95% CI 0.01 to 7.81; participants = 50; studies = 1; $I^2$ = 0%, very low certainty evidence). This suggests that if the rate of all-cause mortality following conventional blood sampling is 4%, the rate of all-cause mortality following the use of devices to monitor glucose levels, subcutaneous would be between 0% and 31%.

**4.3 IVH–severe.** One study reported on severe IVH [85]. We are uncertain of the effect of devices to monitor glucose levels, subcutaneous compared to conventional blood sampling on severe IVH (RR 0.20, 95% CI 0.01 to 3.97; participants = 50; studies = 1; $I^2$ = 0%). This suggests that if the rate of severe IVH following conventional blood sampling is 1%, the rate of severe IVH following the use of devices to monitor glucose levels, subcutaneous would be between 0% and 4%.

**4.4 Late sepsis until hospital discharge.** One study reported on late sepsis until hospital discharge [85]. We are uncertain of the effect of devices to monitor glucose levels, subcutaneous compared to conventional blood sampling on late sepsis until hospital discharge (RR 0.20, 95% CI 0.01 to 3.97; participants = 50; studies = 1; $I^2$ = 0%). This suggests that if the rate of late sepsis until hospital discharge following conventional blood sampling is 1%, the rate of late sepsis until hospital discharge following the use of devices to monitor glucose levels, subcutaneous would be between 0% and 4%.

**4.5 Duration in days of hospital stay.** One study reported on the duration in days of hospital stay [85]. We are uncertain of the effect of devices to monitor glucose levels, subcutaneous compared to conventional blood sampling on the duration in days of hospital stay (presented in median and interquartile range: 51 days (37 to 63 days) in intervention group vs. 46 days (40 to 74 days) in control group (P = 0.59; one study, 50 infants).

**4.6 Number of skin-breaking procedure associated to blood testing, insertion and repositioning of the device.** One study reported on the number of skin-breaking procedures associated with blood testing, insertion and repositioning of the device [87]. The results indicated a potential reduction in the number of blood samples taken in the devices to monitor glucose levels, subcutaneous group compared to the conventional blood sampling group (MD -5.00, 95% CI -7.78 to -2.22, p < 0.001; one study, 48 infants.)

**4.7 Pain during device insertion/use and blood sampling, e.g. heel stick, venipuncture.** The conference abstract of one study reported on pain during device insertion/use and blood sampling, e.g. heel stick, venipuncture [85]. We are uncertain of the effect of devices to monitor glucose levels, subcutaneous compared to conventional blood sampling on pain during device insertion/use and blood sampling, e.g. heel stick, venipuncture (reported on a PIPP-scale presented in median and interquartile range: 4 (3 to 15) in the intervention group vs. 6 (3 to 18) in the control group (no p-value available; one study, seven infants)).

No data were reported for the following outcomes: one-year survival, major neurodevelopmental disability *for children 18 to 24 months*, major neurodevelopmental disability *for children 3 to 5 years of age*, mortality or major neurodevelopmental disability [composite outcome], severe ROP, any ROP, any germinal matrix-IVH, white matter at term-equivalent MRI abnormalities at term equivalent age, BPD/chronic lung disease, necrotizing enterocolitis, volume in mL of blood withdrawn until hospital discharge, volume in mL of blood transfused until hospital discharge, number of blood transfusions until hospital discharge, need for blood transfusions until hospital discharge, concentration of total hemoglobin (Hb), concentration of fetal hemoglobin (HbF), PDA, duration in days of respiratory support, duration in days of supplemental oxygen requirement, each component of the composite outcome 'major neurodevelopmental disability,' poor academic performance at 12 years of age, motor skills problem, behavioral problem, time in minutes to perform the procedure, skin/soft tissue injury associated to blood testing, insertion and repositioning of the device, site infection associated to blood testing, insertion and repositioning of the device, and thrombotic event rates.

## Discussion

This is the first systematic review and meta-analysis to evaluate the benefits and harms of any intervention to minimize blood loss in very preterm infants. Thirty-one trials enrolling 3,759

infants met the review inclusion criteria and were pooled in four different analyses: 25 trials compared delayed cord clamping or cord milking vs. immediate cord clamping or no milking, two trials blood sampling from the umbilical cord or from the placenta vs. blood sampling from the infant, three trials were conducted on subcutaneous devices to monitor glucose levels, and one trial on devices to reintroduce the blood after analysis (Table 1, S2 File).

The 19 trials on delayed cord clamping and the 8 trials on cord milking were analyzed together as they both aim to increase placental transfusion. Placental transfusion largely decreased all-cause neonatal mortality (Fig 4) and, to a minor extent, all-cause mortality during initial hospitalization (Fig 5). These results were likely due to the characteristics of the studies reporting these two outcomes rather than reflecting a major health benefit of transfused blood components in the first four weeks of life. Regarding the other primary outcomes of this review, major neurodevelopmental disability at 24 months and three years of age and the composite outcome death or adverse neurodevelopmental outcome for children at 18 to 24 months were reported in one trial, whereas one-year survival was not reported in any trials. We rated the certainty of evidence (GRADE) as moderate for all-cause neonatal mortality due to imprecision of the estimates, low for all-cause mortality during initial hospitalization due to imprecision of the estimates and inconsistency, and very low for the other primary outcomes due to imprecision of the estimates and indirectness. No relevant differences were identified in following secondary outcomes: ROP, IVH, PDA, NEC and BPD. Placental transfusion resulted in

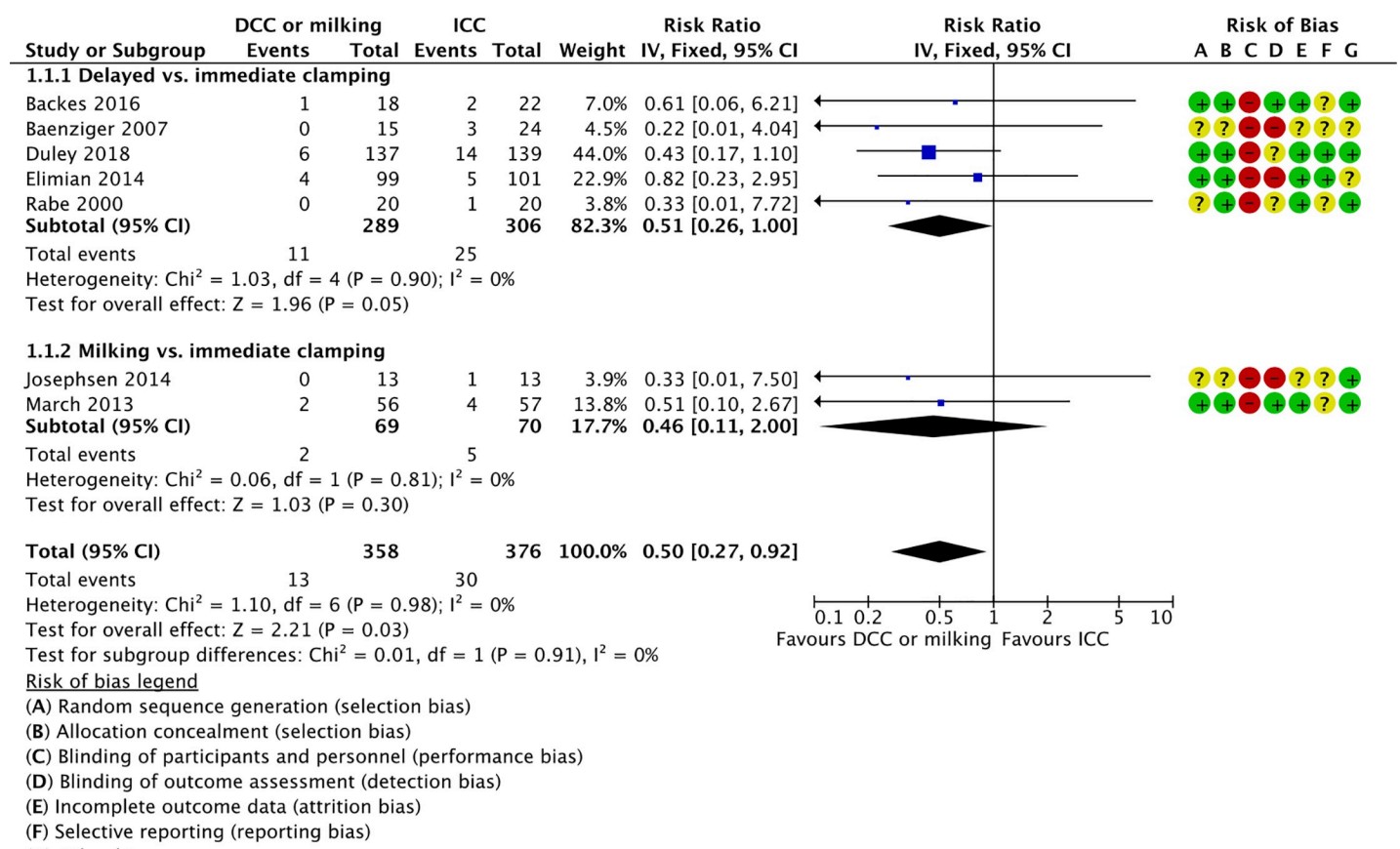

**Fig 4. Forest plot of comparison: 1 delayed cord clamping or milking vs. immediate cord clamping or no milking, outcome: 1.1 all-cause neonatal mortality (first 28 days).**

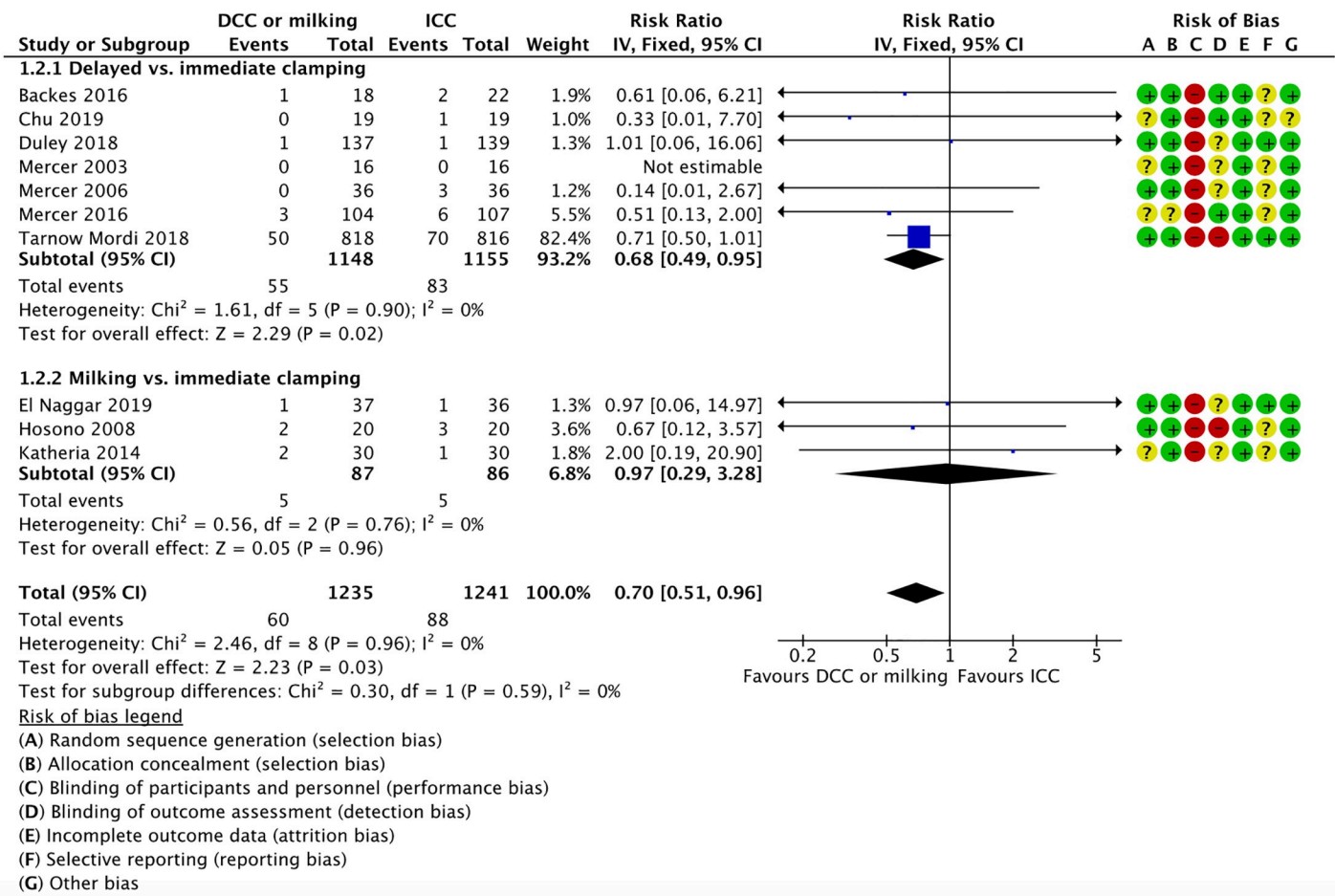

**Fig 5. Forest plot of comparison: 1 delayed cord clamping or milking vs. immediate cord clamping or no milking, outcome: 1.2 all-cause mortality during initial hospitalization.**

a reduced need for blood transfusion and higher concentrations of total hemoglobin in the first week of life (Fig 6), however the certainty of the evidence was very low due to inconsistency and unclear risk of bias in most trials. Our main findings concerning cord management at birth are in line with the 2019 Cochrane review update on delayed cord clamping and cord milking [17], with similar rates in the reduction of mortality up to discharge. Infants were included if under 34-weeks gestational age in the aforementioned review and under 32-weeks' gestational age in ours. In addition, we conducted an overall meta-analysis on delayed cord clamping or cord milking compared to early clamping or no milking. The Cochrane review included a number of relevant subgroup analyses which are not part of this review.

Very few studies were identified on interventions to minimize blood loss following cord management at birth: three trials on glucose monitoring and one trial on a device to reintroduce blood after analysis. No trials were identified on either oxygen or carbon dioxide monitoring, or on micro-methods to sample lower amounts of blood. This finding was unexpected, as some of these devices are largely used in clinical practice and might indeed result in improved outcomes. In the coming years, most of the trials on minimizing blood loss will continue to focus on placental transfusion: only two of the 24 ongoing studies will assess devices to monitor glucose levels or micro-methods for blood analysis; the remaining 22 studies will

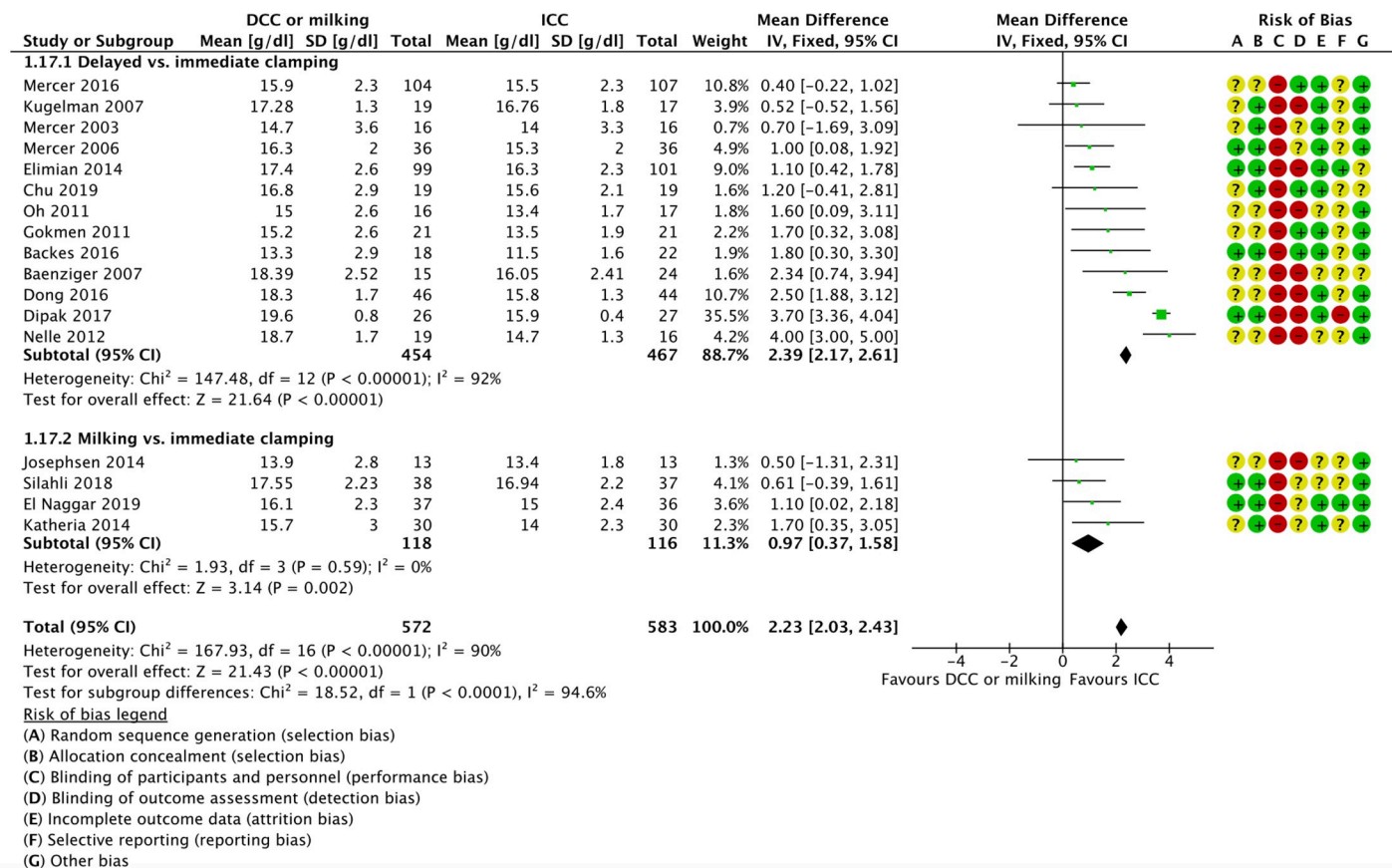

**Fig 6. Forest plot of comparison: 1 delayed cord clamping or milking vs. immediate cord clamping or no milking, outcome: 1.17 concentration of total hemoglobin (Hb)–day 1–7 [g/dl].**

compare delayed cord clamping or cord milking to immediate clamping or no milking (S4 File). We speculate that more awareness of the importance of preserving fetal blood is needed to lead to a change in both clinical management and the establishment of future trials.

Similar systematic reviews focused on one specific type of intervention, such as the review on placental transfusion [17]. The Cochrane review on transcutaneous carbon dioxide monitoring [88] identified one trial which was not included in this review as it was conducted in infants older than 32 weeks of gestational age and during long-distance transport [89]. Our findings on continuous glucose monitoring are in line with the Cochrane review in preterm infants [90].

Overall, the authors of the 31 studies included in our review reported extremely limited data on long-term neurodevelopmental assessment. We could not perform an appropriate *a priori* subgroup analysis to detect differential effects due to the paucity of outcome data amongst the included trials. We were able to summarize the available evidence in a comprehensive way, as we obtained additional information about study design and outcome data from study authors. We assessed the presence of publication bias by creating two funnel plots for the outcomes of highest clinical relevance. Fig 7 corresponds to all-cause neonatal mortality for comparison 1 and Fig 8 corresponds to all-cause mortality during initial hospitalization for comparison 7. However, these analyses should be interpreted with caution due to the paucity of the studies. We used standard methods of the Cochrane Neonatal Review Group in

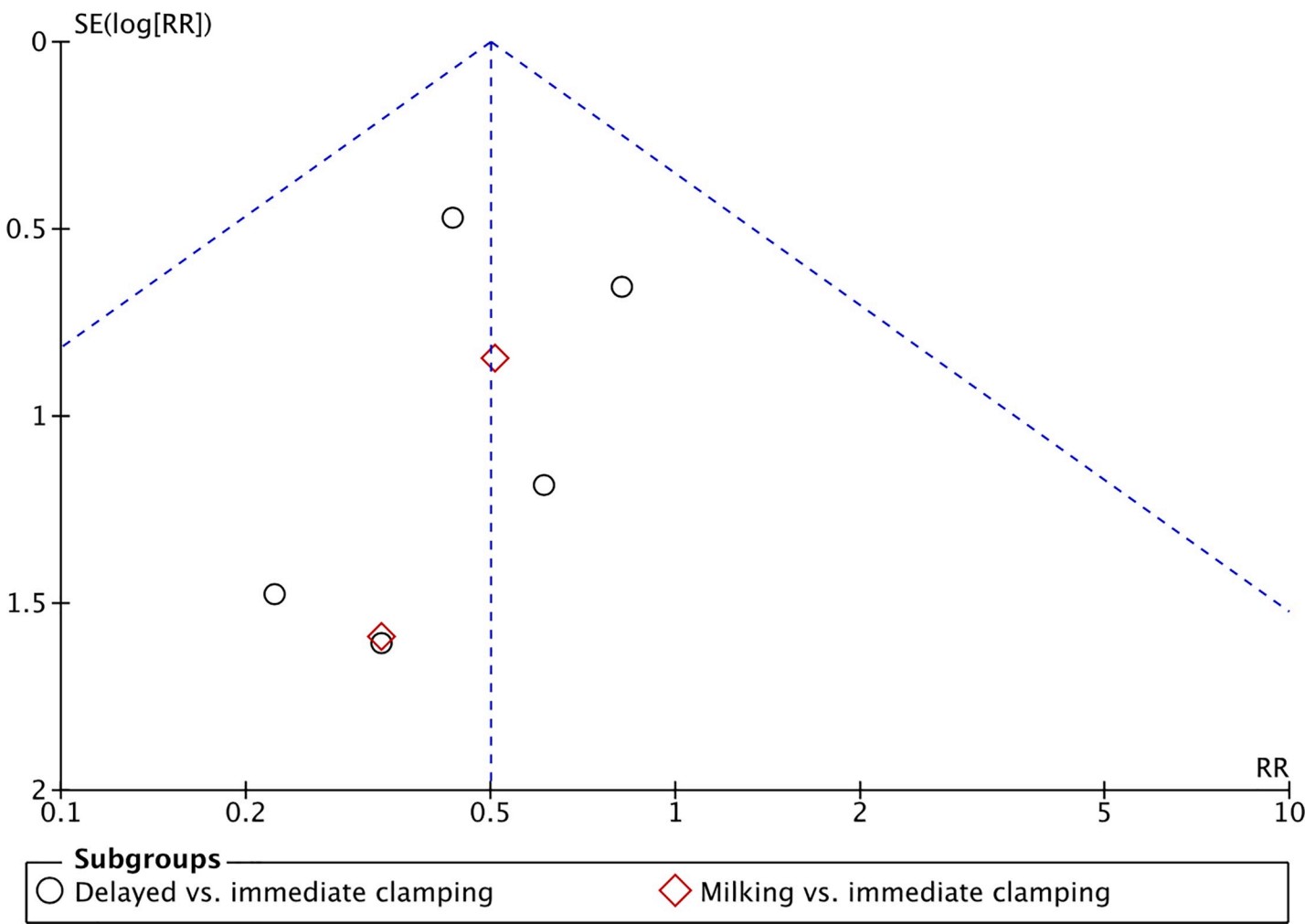

**Fig 7. Funnel plot of comparison: 1 delayed cord clamping or milking vs. immediate cord clamping or no milking, outcome: 1.1 all-cause neonatal mortality (first 28 days).**

conducting this systematic review. One limitation of this review was the search strategy, which was designed to identify the specific interventions we pre-specified in the protocol. However, it is unlikely that the literature search applied to this review missed relevant trials and we are confident that this review is a comprehensive summary of all presently available evidence on minimizing blood loss in very preterm infants.

Pharmacological interventions to increase erythropoiesis were not included in this review as they promote blood production rather than reducing blood loss. We applied no language restriction and succeeded to have trials translated from Mandarin and Hebrew to English. We excluded 80 studies due to the characteristics of the study design, the age of the newborn population (older than 32 weeks of gestational age), the characteristics of the interventions, unclear information, or because they were terminated before completion. We succeeded in obtaining additional information from study authors.

Increasing placental transfusion results in lower neonatal mortality, however there is extremely limited evidence on other outcomes or on devices that reduce blood loss in the first days of life of very preterm infants. Randomizing infants to immediate cord clamping or clamping without milking is not recommended, in line with suggestions from others [17].

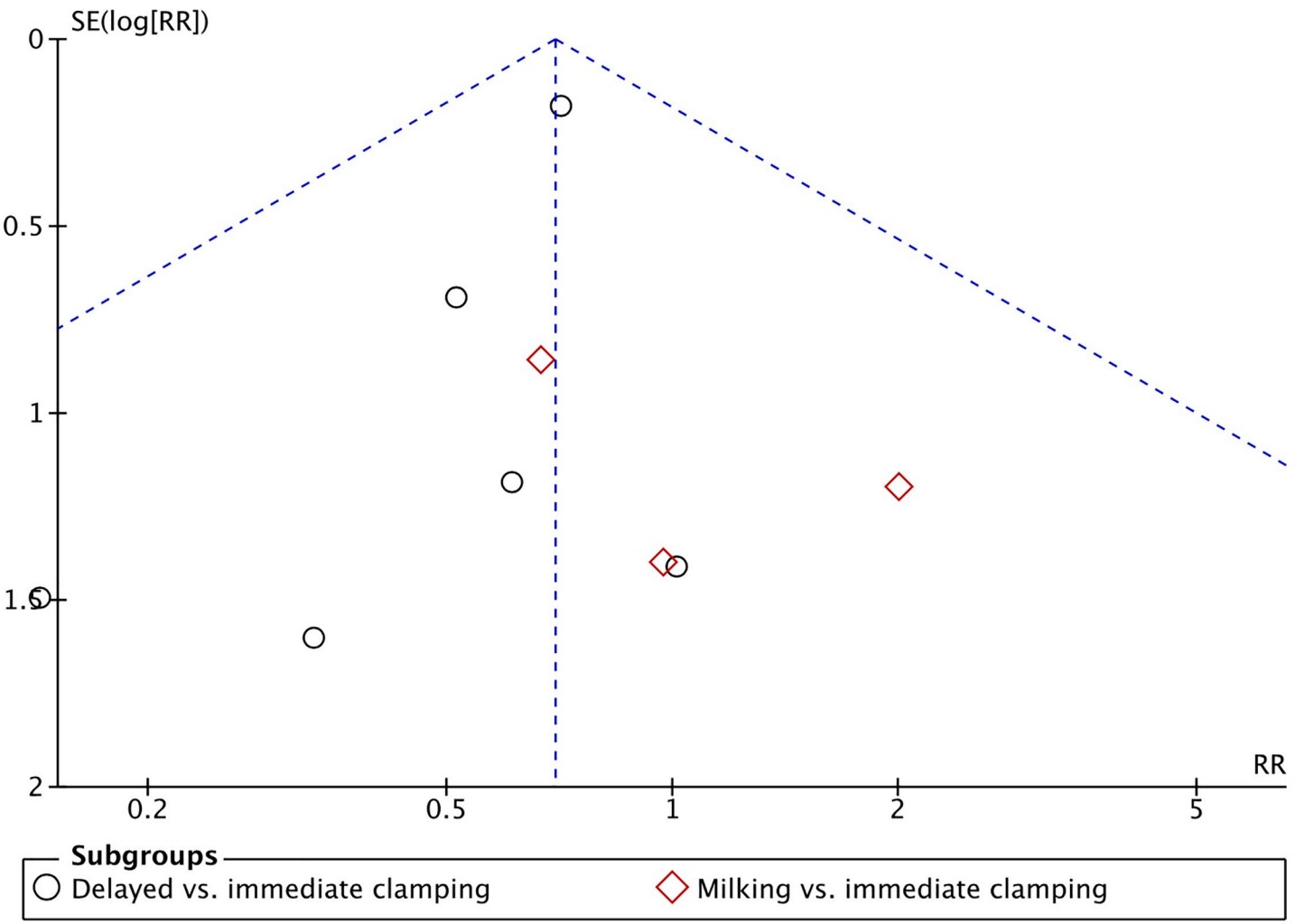

**Fig 8. Funnel plot of comparison: 1 delayed cord clamping or milking vs. immediate cord clamping or no milking, outcome: 1.2 all-cause mortality during initial hospitalization.**

Future trials should consider randomizing infants to different strategies to minimize blood loss following placental transfusion. Moreover, the use of devices to reduce blood loss may be associated with pharmacological interventions to increase erythropoiesis and optimized strategies for blood transfusion. For example, blood collected from the placenta following preterm birth could be used in transfusions to administer higher concentrations of fetal hemoglobin, stem cells, and growth factors. In addition, the specific and sensitivity of interventions to continuously monitor parameters, such as glucose or carbon dioxide levels, should be assessed in systematic reviews of diagnostic test accuracy in very preterm infants. We also reiterate the importance of research for improved awareness of preserving fetal blood following preterm birth.

## Supporting information

**S1 Checklist. PRISMA 2009 checklist.**
(DOC)

**S1 File. Search strategy.**
(DOCX)

**S2 File. Characteristics and risk of bias for each included study.**
(DOCX)

**S3 File. Characteristics of excluded studies.**
(DOCX)

**S4 File. Characteristics of ongoing studies.**
(DOCX)

**S5 File. Characteristics of studies awaiting classification.**
(DOCX)

## Acknowledgments

We have based the Methods section of this review on a standard template used by Cochrane Neonatal (https://neonatal.cochrane.org/).

We would like to thank study authors Dr. Heike Rabe, Dr. Justin Josephsen, Dr. Alfonso Galderisi, Dr. Amir Kugelman, Dr. Anup Katheria, Dr. Himanshu Popat, Dr. Manizheh Mostafa Gharehbaghi, Dr. Melissa March, and Dr. Walid El-Naggar, for proving additional data on their trials.

We would like to thank Dr. Wong Ming Yin (Dental Officer in Ministry of Health, Malaysia) and Dr. Amara Yousef (Poriya Medical Centre, Israel) for the translation of two of the included studies from Mandarin and Hebrew, respectively.

We would like to thank Tamara Kredo (Cochrane South Africa) for her support in the preparation of the summary of findings table.

## Author Contributions

**Conceptualization:** Israel Júnior Borges do Nascimento, Matteo Bruschettini.

**Data curation:** Emma Persad, Greta Sibrecht, Martin Ringsten, Simon Karlelid, Tommy Ulinder, Maria Björklund, Anneliese Arno, Matteo Bruschettini.

**Formal analysis:** Emma Persad, Greta Sibrecht, Martin Ringsten, Simon Karlelid, Matteo Bruschettini.

**Investigation:** Emma Persad, Greta Sibrecht, Martin Ringsten, Simon Karlelid, Olga Romantsik, Tommy Ulinder, Israel Júnior Borges do Nascimento, Matteo Bruschettini.

**Methodology:** Emma Persad, Greta Sibrecht, Martin Ringsten, Simon Karlelid, Olga Romantsik, Tommy Ulinder, Israel Júnior Borges do Nascimento, Matteo Bruschettini.

**Project administration:** Emma Persad, Greta Sibrecht, Maria Björklund, Matteo Bruschettini.

**Resources:** Maria Björklund, Anneliese Arno, Matteo Bruschettini.

**Software:** Anneliese Arno.

**Supervision:** Matteo Bruschettini.

**Validation:** Martin Ringsten, Matteo Bruschettini.

**Visualization:** Matteo Bruschettini.

**Writing – original draft:** Emma Persad, Greta Sibrecht, Martin Ringsten, Simon Karlelid, Olga Romantsik, Tommy Ulinder, Israel Júnior Borges do Nascimento, Matteo Bruschettini.

**Writing – review & editing:** Emma Persad, Greta Sibrecht, Olga Romantsik, Tommy Ulinder, Matteo Bruschettini.

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
