## [Decision Letter · Decision Letter 0]

9 Dec 2020

PONE-D-20-31636

Interventions to minimize blood loss in very preterm infants – a systematic review and meta-analysis

PLOS ONE

Dear Dr. Bruschettini,

Thank you for submitting your manuscript to PLOS ONE. After careful consideration, we feel that it has merit but does not fully meet PLOS ONE’s publication criteria as it currently stands. Therefore, we invite you to submit a revised version of the manuscript that addresses the minor points raised during the review process.

We look forward to receiving your revised manuscript.

Kind regards,

Olivier Baud, MD, PhD

Academic Editor

PLOS ONE

Journal Requirements:

2. Please include your tables as part of your main manuscript and remove the individual files. Please note that supplementary tables should be uploaded as separate "supporting information" files.

Reviewers' comments:

Reviewer's Responses to Questions

**Comments to the Author**

1. Is the manuscript technically sound, and do the data support the conclusions?

Reviewer #1: Yes

2. Has the statistical analysis been performed appropriately and rigorously? 

Reviewer #1: Yes

3. Have the authors made all data underlying the findings in their manuscript fully available?

Reviewer #1: Yes

4. Is the manuscript presented in an intelligible fashion and written in standard English?

Reviewer #1: Yes

5. Review Comments to the Author

Reviewer #1: The authors present a well written systematic review and meta-analysis manuscript focusing on the potential influence of blood loss on common prematurity co-morbidities and interventions to prevent blood loss in the infants from birth. This is an important review and one of interest in many neonatologists.

However, I have some minor issues and comments:

Minor issues:

1. Section "Results", page 17, sentence 414-415 : “Three studies were relevant for the comparison blood sampling from the umbilical cord or from the placenta vs. blood sampling from the infant”: The studies using umbilical cord or from the placenta vs. blood sampling from the infant are two and not three. There is an inconsistency here, authors must verify it throughout the article.

2. Section "Results", page 17, sentence 417-418 :, “two studies were relevant for the comparison devices to monitor glucose levels, subcutaneous vs. conventional blood sampling“: The studies using u devices to monitor glucose levels, subcutaneous vs. conventional blood sampling are three and not two. There is an inconsistency here, authors must verify it throughout the article.

Minor comments:

1. Section "Methods": Did the authors think that their search strategy was good enough to cover all the relevant studies? Why pre-specified the specific interventions? They could have missed relevant trials for the systematic review. Add maybe a suggested optimisation for this search strategy in the discussion section

2. Section "Methods": Why did the authors decide to apply no language restrictions to the search strategy? How did they handle the translations without any error or bias?

3. Section "Methods": Why did the authors decide to use the I2 statistic heterogeneity measure and not others, like Q statistic, H2 statistic, R2 statistic, etc.?

4. Section “Discussion”, page 50, sentence 1216: Concerning the figures 7 and 8, it’s not clear in this section to what correspond each figure. Authors must clarify the methods used for these funnel plots. These methods are not presented in the “method” section, including them will improve the consistency of the manuscript.

5. Section “figures”, Funnel plots, fig 7 and 8: Authors should label the difference between the two funnel plots in the figures’ section as they are almost identical.

6. PLOS authors have the option to publish the peer review history of their article (what does this mean?). If published, this will include your full peer review and any attached files.

Reviewer #1: No

---

## [Author Response · Author response to Decision Letter 0]

15 Dec 2020

The response letter is attached as a PDF.

Many thanks to Reviewers and the Editor for the much useful feedback.

Best wishes, Matteo Bruschettini

---

## [Decision Letter · Decision Letter 1]

18 Jan 2021

Interventions to minimize blood loss in very preterm infants – a systematic review and meta-analysis

PONE-D-20-31636R1

Dear Dr. Bruschettini,

We’re pleased to inform you that your manuscript has been judged scientifically suitable for publication and will be formally accepted for publication once it meets all outstanding technical requirements.

Kind regards,

Olivier Baud, MD, PhD

Academic Editor

PLOS ONE

Additional Editor Comments (optional):

Reviewers' comments:

Reviewer's Responses to Questions

**Comments to the Author**

1. If the authors have adequately addressed your comments raised in a previous round of review and you feel that this manuscript is now acceptable for publication, you may indicate that here to bypass the “Comments to the Author” section, enter your conflict of interest statement in the “Confidential to Editor” section, and submit your "Accept" recommendation.

Reviewer #1: All comments have been addressed

2. Is the manuscript technically sound, and do the data support the conclusions?

Reviewer #1: Yes

3. Has the statistical analysis been performed appropriately and rigorously? 

Reviewer #1: Yes

4. Have the authors made all data underlying the findings in their manuscript fully available?

Reviewer #1: Yes

5. Is the manuscript presented in an intelligible fashion and written in standard English?

Reviewer #1: Yes

6. Review Comments to the Author

Reviewer #1: I would like to thank the authors for addressing my comments and for all the modifications in the manuscript.

7. PLOS authors have the option to publish the peer review history of their article (what does this mean?). If published, this will include your full peer review and any attached files.

Reviewer #1: No

---

## [Editor Report · Acceptance letter]

22 Jan 2021

PONE-D-20-31636R1 

Interventions to minimize blood loss in very preterm infants – a systematic review and meta-analysis 

Dear Dr. Bruschettini:

I'm pleased to inform you that your manuscript has been deemed suitable for publication in PLOS ONE. Congratulations! Your manuscript is now with our production department. 

Kind regards, 

on behalf of

Pr. Olivier Baud 

Academic Editor

PLOS ONE